# Paralogous synthetic lethality underlies genetic dependencies of the cancer-mutated gene *STAG2*

Melanie L Bailey[1], David Tieu[2], Andrea Habsid[2], Amy Hin Yan Tong[2], Katherine Chan[2], Jason Moffat[2,3,4], Philip Hieter[1]

***STAG2*, a component of the mitotically essential cohesin complex, is highly mutated in several different tumour types, including glioblastoma and bladder cancer. Whereas cohesin has roles in many cancer-related pathways, such as chromosome instability, DNA repair and gene expression, the complex nature of cohesin function has made it difficult to determine how *STAG2* loss might either promote tumorigenesis or be leveraged therapeutically across divergent cancer types. Here, we have performed whole-genome CRISPR-Cas9 screens for *STAG2*-dependent genetic interactions in three distinct cellular backgrounds. Surprisingly, *STAG1*, the paralog of *STAG2*, was the only negative genetic interaction that was shared across all three backgrounds. We also uncovered a paralogous synthetic lethal mechanism behind a genetic interaction between *STAG2* and the iron regulatory gene *IREB2*. Finally, investigation of an unusually strong context-dependent genetic interaction in HAP1 cells revealed factors that could be important for alleviating cohesin loading stress. Together, our results reveal new facets of STAG2 and cohesin function across a variety of genetic contexts.**

## Introduction

Cohesin is a well-conserved structural ring complex that physically tethers two DNA segments to ensure proper chromatin organization and function. In more complex organisms like humans, cohesin is involved in many diverse cellular functions including sister chromatid cohesion, DNA repair, replication fork restart, transcription, and promotion of topologically associating domains (Losada, 2014; Waldman, 2020). As a multisubunit complex, mitotic cohesin consists of three core ring components, SMC1A, SMC3, and RAD21, as well as several accessory factors including one of two SCC3 paralogs, either STAG1 or STAG2 (Uhlmann, 2016). Although these two paralogs are thought to be mainly interchangeable in the complex, more recent data have indicated separate roles for cohesin-STAG1 and cohesin-STAG2 in a small number of functions (Kong et al, 2014; Daniloski & Smith, 2017; Kojic et al, 2018; Casa et al, 2020).

In cancer, the *STAG2* gene is more highly mutated than any other cohesin component, including *STAG1* (Hill et al, 2016; Waldman, 2020). In fact, *STAG2* is 1 of only 12 genes significantly recurrently mutated in four or more tumour types (Lawrence et al, 2014) although how STAG2 loss-of-function might promote tumourigenesis remains unclear. Although earlier studies could find no clear correlation between *STAG2* mutation and genome instability (Balbás-Martínez et al, 2013; Hill et al, 2016; Benedict et al, 2020), loss of STAG2 has more recently been linked to an increase in telomere recombination and a decrease in cell type-specific transcription, both of which could have implications in cellular transformation (Mullenders et al, 2015; Galeev et al, 2016; Daniloski & Smith, 2017; Kojic et al, 2018). Loss of *STAG2* has also been shown to be synthetic lethal with its paralog *STAG1* (Benedetti et al, 2017; van der Lelij et al, 2017, 2020; Liu et al, 2018; Viny et al, 2019).

The identification of genetic interactions for a gene of interest, such as a cancer gene, can provide important functional and therapeutic information (O'Neil et al, 2017; Mair et al, 2019). Fortunately, the discovery of CRISPR-Cas9 has made it technically feasible to screen for and test candidate genetic interactions in mammalian systems (Hart et al, 2015; Wong et al, 2016; Behan et al, 2019). In recent years, several groups have proposed a need to incorporate a deeper understanding of the context-dependency of a candidate genetic interaction, specifically the cell type(s) or cell line background(s) where the genetic interaction can be found (Ryan et al, 2018; Shen & Ideker, 2018). A genetic interaction target observed in many cell backgrounds (i.e., a more context-independent target) has the advantage of not only being a potential therapeutic target across multiple cancer types, but also has a higher likelihood of being more robust in the context of tumour heterogeneity. So far, mechanisms behind context-dependent genetic interactions remain understudied, but recent studies have noted that interactions involving paralogous synthetic lethality are often found in more than one cellular background (Tsherniak et al, 2017; Dede et al, 2020; Gonatopoulos-Pournatzis et al, 2020; Lord et al, 2020). Paralogous synthetic lethality is a genetic interaction that occurs when the loss of one paralog in an essential complex or pathway is buffered by the presence of a second paralog (Muller et al, 2012; D'Antonio et al, 2013; Tsherniak et al, 2017).

Given the importance of STAG2 in cohesin function and its high rate of mutation in diverse types of cancer (Lawrence et al, 2014;

[1]Michael Smith Laboratories, University of British Columbia, Vancouver, Canada    [2]Donnelly Centre, University of Toronto, Toronto, Canada    [3]Department of Molecular Genetics, University of Toronto, Toronto, Canada    [4]Institute of Biomedical Engineering, University of Toronto, Toronto, Canada

Correspondence: hieter@msl.ubc.ca

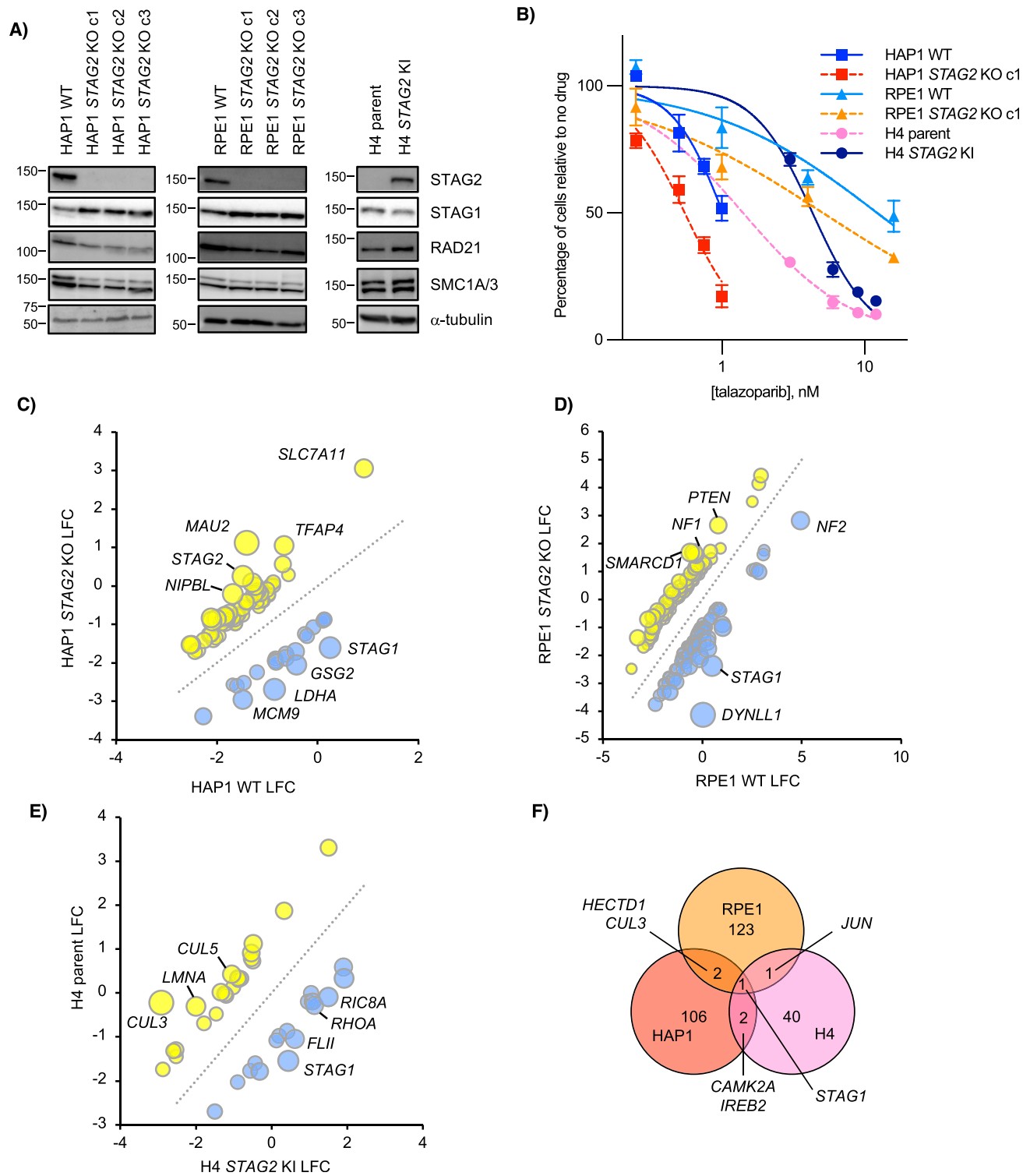

**Figure 1. Whole-genome CRISPR-Cas9 knockout screen in cell lines with and without *STAG2*.**
**(A)** Western blots of *STAG2*+ and *STAG2*– clones in three different backgrounds. Note that whereas HAP1 and RPE1 parents contain wild-type *STAG2*, H4 glioblastoma parent cells contain an endogenous *STAG2* insertion that leads to protein truncation which is corrected in the knock-in (KI) cell line. **(B)** Effect of the PARP inhibitor talazoparib on select *STAG2*+ and *STAG2*– cells in three different backgrounds. Each line was normalized to a no drug control. Data were fitted with a non-linear IC$_{50}$ curve in GraphPad Prism. **(C, D, E)** Comparison of *STAG2*+ log$_2$-fold change scores and *STAG2*– log$_2$-fold change scores in (C) HAP1, (D) RPE1, and (E) H4 backgrounds at FDR < 0.2. Candidate negative genetic interactions are in blue and candidate positive genetic interactions are in yellow. **(F)** Overlap of candidate *STAG2* negative genetic interactions at a cut-off of qGI < –0.6 for HAP1 and residual < –1 for RPE1 and H4. % overlap at this cut-off is 2.18%.

▶▶▶ Life Science Alliance

Hill et al, 2016; Waldman, 2020), we performed genome-wide CRISPR-Cas9 knockout screens in three different genetic backgrounds to look for *STAG2*-dependent genetic interactions that could provide new insights into cohesin-STAG2 function and/or *STAG2*-based therapeutic strategies. Our screens confirmed *STAG1* as the sole, strong context-independent negative genetic interaction for *STAG2*. We also found a previously unknown genetic interaction between *STAG2* and an iron regulatory gene and a unique *STAG2*-dependent context in which the cohesin loading complex is no longer needed for cell viability. Collectively, our results highlight the multifunctional nature of cohesin-STAG2 and suggest that STAG1 is the only context-independent "loss-of-function" therapeutic target.

# Results

### CRISPR-Cas9 screening of *STAG2*-positive and *STAG2*-negative cell lines

To screen for genetic interactions with *STAG2*, we first obtained a series of *STAG2*-positive (*STAG2+*) and *STAG2*-negative (*STAG2−*) isogenic clones in three distinct cell line backgrounds. For HAP1 and RPE1 lines, we generated *STAG2* KO clones using one of two single guide RNAs (sgRNA) that targeted *STAG2* exonic DNA (Fig S1A and Table S1). Both HAP1 and RPE1 lines have been screened with CRISPR-Cas9 previously (Hart et al, 2015, 2017; Brown et al, 2019; Aregger et al, 2020), which we felt would be helpful in analyzing results from our isogenic *STAG2* screens. Knockout of *STAG2* in these two lines produced mainly small indels and coding frameshifts that led to early protein truncation (Table S1). We also obtained H4 parent cells that contain an endogenous truncating *STAG2* mutation that has been corrected ex vivo in the H4 *STAG2* knock-in (KI) line. These H4 lines have been described previously (Solomon et al, 2011) and were chosen because the parent represented a context in which the *STAG2* mutation occurred and was adapted to in a tumour rather than being CRISPR generated. Western blots of all three cell line contexts confirmed the absence of STAG2 protein in *STAG2−* lines (Fig 1A). Our *STAG2−* lines also often showed compensatory up-regulation of STAG1 at the protein level (Fig 1A), a phenomenon that has been reported previously with RNAi (Kong et al, 2014).

As our HAP1 and RPE1 *STAG2* KO cell lines were newly derived, we sought to further characterize them before performing whole-genome screening. To do this, we subjected parent and derived lines to a panel of 11 genotoxic and chemical stresses, some of which had shown *STAG2*-dependent responses previously (McLellan et al, 2012; Bailey et al, 2014; Mondal et al, 2019). Both HAP1 and RPE1 cells showed several *STAG2*-dependent responses, but these were often cell background-specific (Fig S1B). Only poly (ADP-ribose) polymerase (PARP) inhibitors showed a higher sensitivity in *STAG2*-depleted cells across all backgrounds (Figs 1B and S1B) demonstrating that our HAP1 and RPE1 *STAG2* KO cell lines aligned with previous reports where *STAG2* status was found to contribute to PARP inhibitor sensitivity (McLellan et al, 2012; Bailey et al, 2014; Mondal et al, 2019).

After genetic and chemical characterization, we undertook genome-wide CRISPR-Cas9 knockout screens of *STAG2+* and *STAG2−* lines in all

three cell backgrounds using the Toronto Knockout (TKO) v3 CRISPR library (Hart et al, 2017; Aregger et al, 2019) (Fig S1C). All screens showed dropout of a reference set of core essential genes compared with a reference set of non-essential genes (Fig S1D and Table S2) (Hart et al, 2014, 2017). In addition, there was generally good correlation of $\log_2$(aggregate sgRNA fold-change of end/initial reference time points) between different screens (Fig S1E). *STAG2*-dependent genetic interaction scores were calculated for each cell line background (see the Materials and Methods section; Tables S3–5) and candidate negative and positive genetic interactions at FDR < 0.2 for each screen are shown in Fig 1C–E. Genetic interaction overlap between the three cell line backgrounds is shown with cut-offs for genetic interaction strength (Fig 1F) and statistical significance (Fig S1F). In both cases, only *STAG1* shared a negative interaction in all three lines (Figs 1F and S1F) and few negative genetic interactions were observed with *STAG2* across multiple genetic backgrounds, suggesting that the loss of *STAG2* is highly buffered by *STAG1* (Figs 1F and S1F).

### *STAG1/STAG2* is a context-independent paralogous synthetic lethal interaction

*STAG1/STAG2* is an example of paralogous synthetic lethality where the two genes involved are two interchangeable subunits in an essential protein complex. In addition to our primary screens (Fig 1C–E), *STAG1* and *STAG2* have also recently been identified as a negative genetic interaction in several other backgrounds, including both H4 and RPE1 *TP53* knockouts (Benedetti et al, 2017; van der Lelij et al, 2017, 2020; Liu et al, 2018; Mondal et al, 2019; Viny et al, 2019). To study the mechanism of this interaction further, we used CRISPR interference (CRISPRi) to knockdown *STAG1* in wild-type or *STAG2* KO lines (Fig S2A). Consistent with our primary screens, knockdown of *STAG1* resulted in a preferential decrease in cell growth only in *STAG2*-deficient lines in all cell backgrounds (Fig 2A–C). Previous studies have shown that cells depleted of both *STAG1* and *STAG2* have a variety of mitotic phenotypes including defective sister chromatid cohesion, defective metaphase alignment and an increased mitotic index (van der Lelij et al, 2017, 2020; Liu et al, 2018). Although cell cycle distribution changes varied in our *STAG2*-deficient/*STAG1*-depleted backgrounds (Fig S2B), knockdown of *STAG1* in *STAG2−* cell lines always resulted in an increased mitotic index as measured by an increase in phospho-serine 10 Histone H3-positive (pH3+) cells (Figs 2D–F and S2B). This suggests a shared mechanism for the *STAG1/STAG2* interaction regardless of cell background. Interestingly, HAP1 cells depleted of both STAG components showed a much higher increase in sub-G1 cells than the other cell lines (Fig S2B) which we speculate could be due to mitotic defects and/or *TP53* mutations present in this background (Haarhuis et al, 2013; Yaguchi et al, 2018).

To determine whether the dependence of *STAG2*-mutated cells on *STAG1* was a generalizable interaction beyond our three paired backgrounds, we examined the AVANA CRISPR Cancer Dependency Map (DepMap) dataset (Tsherniak et al, 2017; Behan et al, 2019). These data clearly showed a lower *STAG1* gene effect score in cell lines with deleterious or damaging *STAG2* mutations compared with those containing wild-type or other *STAG2* mutations (Fig S2C). This suggests that, compared with *STAG2+* cells, *STAG2−* cells are more dependent on *STAG1* on a large, context-independent scale, a finding

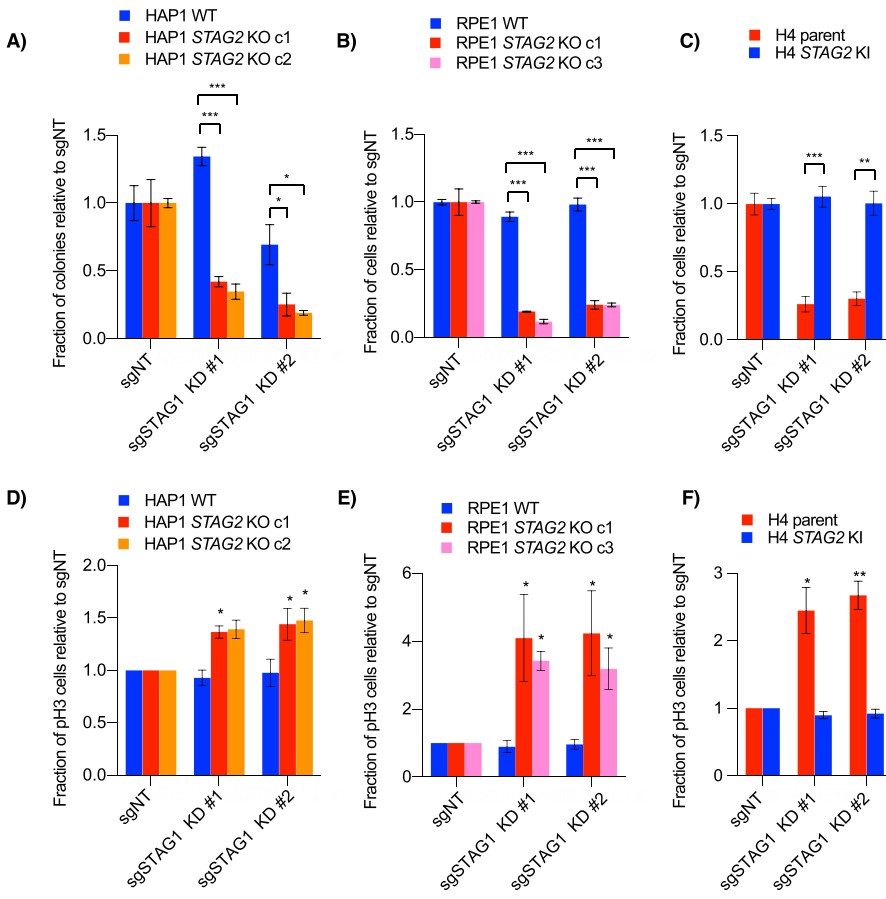

**Figure 2. *STAG1* knockdown in *STAG2* KO cells.**
**(A, B, C)** Cell growth after infection of non-targeting or *STAG1* knockdown sgRNAs in (A) HAP1, (B) RPE1, or (C) H4 cell lines stably expressing dCas9-KRAB. For the HAP1 lines, proliferation was determined by clonogenic assay. For H4 glioblastoma and RPE1 cells, cell growth was determined by nuclei counting. **(D, E, F)** Relative number of mitotic cells as analyzed by phospho-Ser10 Histone H3 (pH3) staining and flow cytometry. Data in (D, E, F) represent the average of three independent experiments. *$P < 0.05$, **$P < 0.005$, ***$P < 0.0005$ in a either a two-tailed, Welch's unpaired (A, B, C) or a one-tailed, paired (D, E, F) $t$ test.

consistent with previous knockdown of *STAG1* in bladder cancer and Ewing's cell line panels (van der Lelij et al, 2017).

## A genetic interaction between *STAG2* and the iron regulatory gene *IREB2*

Whereas *STAG1* was the only negative genetic interaction shared in all three cell backgrounds, several other gene candidates were found in two out of three lines and we wondered if we could learn more about STAG2 function and genetic interaction context-dependency by further characterizing one of these, namely, the *STAG2*/*IREB2* interaction observed in the cancer-derived HAP1 leukemia and H4 glioblastoma cell lines.

In normal cells, iron levels must be tightly controlled (Fig 3A). Sufficient iron levels are needed for cell essential processes such as metabolism and DNA synthesis; however, excess iron can promote the formation of free radicals that contribute to cellular damage and death (Katsarou & Pantopoulos, 2020). *IREB2* (protein name IRP2) and its paralog *ACO1* control the iron regulatory response by directly responding to changes in intracellular iron levels. As iron levels increase, the IRP2 protein is degraded, whereas ACO1 protein levels remain unaffected, but the protein undergoes a conformational change to an aconitase (Fig 3A) (Kühn, 2015). Conversely, when iron levels in the cell are low, both iron response protein (IRP) proteins can bind iron-responsive element (IRE)–

containing mRNAs which may either increase in abundance or decrease expression depending on IRE locations and mRNA function (Kühn, 2015). Interestingly, increases in intracellular iron have been observed in many different tumour types (Torti & Torti, 2013), and we wondered if there might be a connection between STAG2 loss-of-function and iron dysregulation given the potential *STAG2*/*IREB2* genetic interaction in HAP1 and H4 cells, the two transformed cell line contexts (Fig 1F).

Cell growth assays confirmed that knockout of *IREB2* was more detrimental in *STAG2*− cells than *STAG2*+ cells in both the HAP1 and H4 contexts (Figs 3B–D and S3A–D). The same negative genetic interaction was also observed in a 42 MGBA parent/*STAG2* KI cell line pair which, like the H4 context, contains a naturally occurring truncating mutation in the parent line which is corrected in the KI line (Solomon et al, 2011), as well as a HCT116 WT/*STAG2* KO cell line pair (Fig S3E–G). Interestingly, we did not see a negative genetic interaction between *STAG2* and *IREB2* in RPE1 cell lines (Figs 3E and S3H), consistent with the primary screen data. We note that in all our backgrounds the *STAG2*/*IREB2* interaction appears more moderate than *STAG1*/*STAG2*. Furthermore, DepMap screening data also showed more modest *STAG2*-dependent differences in the *IREB2* gene effect score in various lineages (Fig S3I). Therefore, we conclude that *STAG2*/*IREB2* is a modest negative genetic interaction in four out of five of our *STAG2* isogenic contexts.

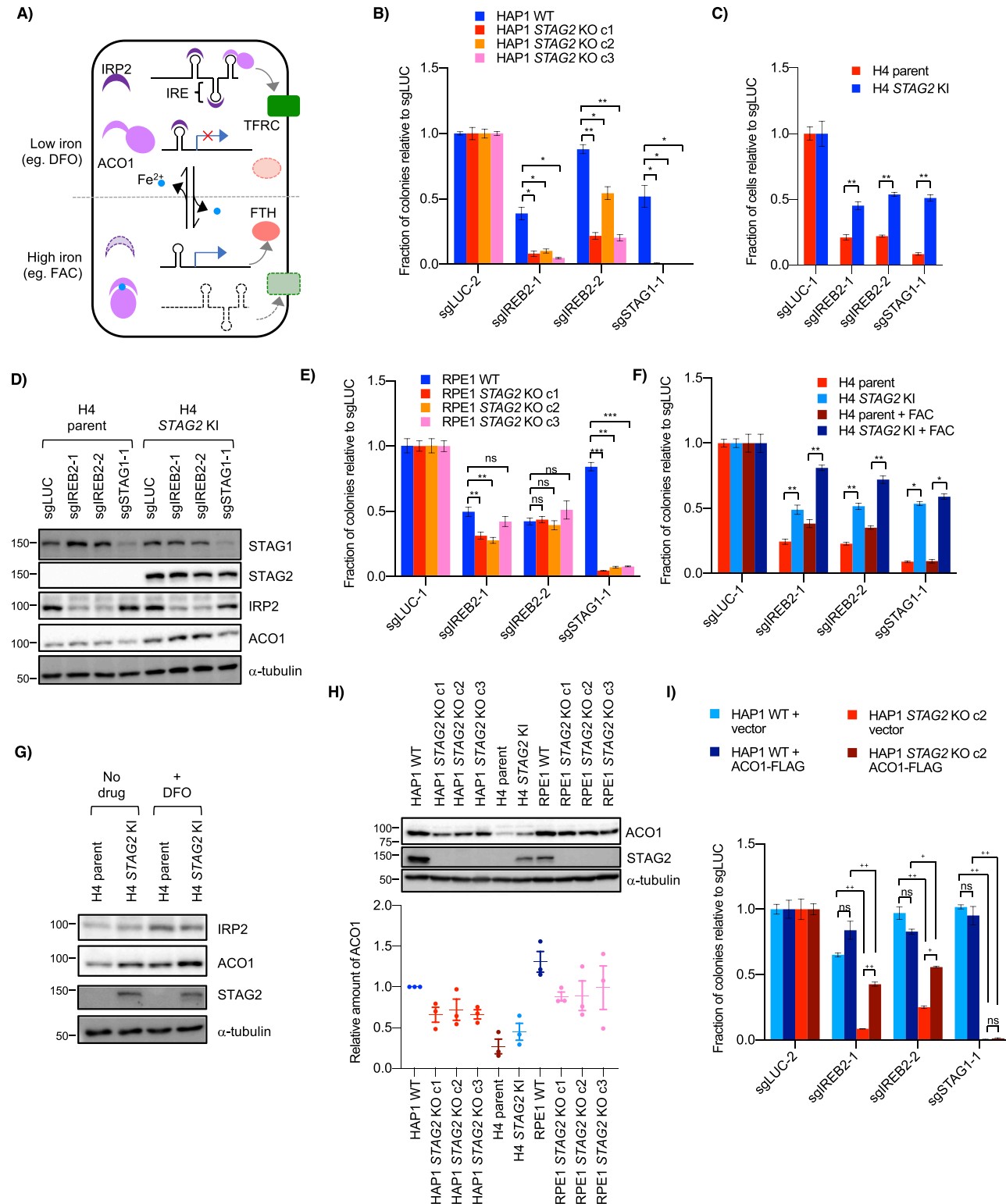

**Figure 3. Genetic interaction between *STAG2* and *IREB2*.**
**(A)** Schematic of ACO1 and IRP2 function in cells. In low-iron conditions (e.g., DFO), ACO1 is in an RNA-binding form and IRP2 (the gene product of *IREB2*) is stabilized. Both bind iron responsive elements in target mRNAs and either stabilize them (as in the case of the iron importer TFRC) or block their translation (as with the ferritin component FTH). In high iron conditions (e.g., FAC), the IRP proteins are not needed and ACO1 is converted to an aconitase form, whereas IRP2 is degraded which destabilizes the mRNA of TFRC and allows translation of FTH. **(B)** Growth of HAP1 WT and *STAG2* KO cells after transfection of control or *IREB2* sgRNA as determined by clonogenic assay. **(C)** Interaction between *STAG2* and *IREB2* KO in H4. Cells were infected with control and *IREB2* sgRNA before being normalized and re-plated for growth

As *IREB2* is a master iron regulator in the cell, we tested whether adding excess iron affected the interaction of *STAG2* and *IREB2*. We found that, compared with standard media, there was no change in the strength of the negative genetic interaction between *IREB2* and *STAG2* when ferric ammonium citrate (FAC) was added (Fig 3F) although iron addition did partially rescue proliferation defects after *IREB2* knockout in both H4 parent and *STAG2* KI lines indicating FAC treatment was partially effective. The genetic interaction was also unaffected when a low level of deferoxamine (DFO) was added (Fig S3J). DFO is an iron chelator that has been used clinically to treat excess iron burden (Kontoghiorghe & Kontoghiorghes, 2016).

The above results suggest that *STAG2–* cells are not more dependent on *IREB2* for proliferation because of an upstream iron-related stress or a down-stream iron-dependent dysregulation. We next considered how STAG2 loss-of-function affected IRP IRP2 and its paralog ACO1. Western blots of *STAG2+* and *STAG2–* cells showed similar levels of IRP2 even after altering IRP2 levels with DFO (Fig 3G). There was, however, a detectable decrease in the IRP2 paralog ACO1 in the *STAG2–* lines compared with the *STAG2+* lines across multiple conditions and backgrounds (Fig 3D, G, and H). Quantification of ACO1 levels in HAP1, H4, and RPE1 *STAG2+/–* backgrounds showed absolute levels of ACO1 differed across the three backgrounds (Fig 3H). Interestingly, RPE1 cells had the highest ACO1 levels and also did not demonstrate a *STAG2/IREB2*–negative genetic interaction. As the DepMap CRISPR screening data suggested that lower ACO1 levels can affect the proliferative response of *IREB2* KO, at least up to a certain point (Fig S3K), we speculated that there was a threshold expression below which levels of ACO1 are not sufficient to fully buffer loss of *IREB2*.

To determine if higher ACO1 levels affect the strength of the *STAG2/IREB2* interaction, we co-expressed ACO1-FLAG and *IREB2* sgRNAs in HAP1 *STAG2+/–* cells. Exogenous ACO1 expression partially rescued the proliferative defect in *STAG2* KO cells, but did not have a significant effect in WT cells (Figs 3I and S3L). This suggests that increasing expression of ACO1 can mitigate the strength of the *STAG2/IREB2* interaction and provides an explanation for why RPE1 cells, which have higher endogenous levels of ACO1, do not show a significant *STAG2/IREB2* interaction. It also suggests that decreases in ACO1 expression in *STAG2–* cells can make them more susceptible to loss of the other IRP paralog when absolute ACO1 levels are below a threshold.

To further investigate the effects of *STAG2* loss on iron regulation, we treated *STAG2+* and *STAG2–* cell lines with the iron chelator DFO. Across multiple backgrounds, we found no consistent *STAG2*-dependent proliferation response to DFO after either acute or chronic treatment with the chelator, providing no obvious link between *STAG2* status and growth in low iron conditions (Figs 4A–C and S4A–C). We also looked at levels of two IRP targets: TFRC/TFR1, an iron importer that increases in low iron and ferritin heavy chain (FTH), a subunit of

the ferritin storage complex, that increases in high iron (Torti & Torti, 2013). Western blots showed increased stabilization of FTH in *STAG2–* cells compared with *STAG2+* cells after iron addition, but no *STAG2*-dependent difference in TFRC (Figs 4D and S4D). Although *ACO1* and *IREB2* are paralogs and largely functionally redundant, molecular studies have shown preferential binding affinities to several IRE sequences and target regulation which may partly explain this result (Wang et al, 2007; Kühn, 2015). As well, *IREB2* is thought to be the more important regulator for iron response in mice and in human cells (Meyron-Holtz et al, 2004; Wang et al, 2007), which may make ACO1-dependent responses in *STAG2* KO cells more difficult to observe when *IREB2* is present. We therefore cannot confirm a link between *STAG2* KO and iron response and suggest the mechanism behind the *STAG2/IREB2* genetic interaction occurs at the paralog level in isogenic lines. In *STAG2* KO lines, IRP2 can likely compensate for decreased ACO1 levels. It is only after *IREB2* is depleted and *ACO1* becomes the dominant paralog that cells are more susceptible to *STAG2*-dependent regulation.

## A unique *STAG2*-dependent positive genetic interaction with cohesin loaders

Although the original goal of our *STAG2* screen was to look for negative genetic interactions common to multiple backgrounds, we also observed an unexpected positive candidate interaction in the HAP1 primary screen between *STAG2* and both subunits of the cohesin loader (Fig 1C). In humans, the heterodimeric cohesin loader complex, composed of NIPBL and MAU2, loads cohesin onto DNA, typically in regions associated with promoters and active transcription (Liu et al, 2009; Newkirk et al, 2017). Although the cohesin loader genes are essential in *Saccharomyces cerevisiae*, more recent studies in vitro and in mammalian cells have suggested a role for both STAG1 and STAG2 in the chromatin localization of cohesin, either through binding of the insulator protein, CTCF, or by binding DNA directly (Rubio et al, 2008; Xiao et al, 2011; Bisht et al, 2013; Countryman et al, 2018; Kojic et al, 2018; Pherson et al, 2019). It was surprising then, that *NIPBL* and *MAU2* were identified as strong candidate positive genetic interactions in the HAP1 background (Fig 1C). This would suggest that after *STAG2* is lost, the loaders become less essential in this background.

To verify the genetic interaction, we used clonogenic assays and confirmed that knockout of either *MAU2* or *NIPBL* resulted in a positive interaction with *STAG2* KO in HAP1 but not RPE1 or H4 cell line backgrounds (Figs 5A and B and S5A and B). Using HAP1 *STAG1* clonal knockout cell lines, we observed a similar suppressive interaction between *MAU2* and *STAG1* (Fig S5C). Double transfection of sgRNAs for *STAG2* and *MAU2* into HAP1 WT cells also resulted in a positive interaction (Fig S5D). Together, these data confirmed a

---

assessment. Cell number was determined by nuclei counting. **(D)** Levels of iron regulatory proteins in H4 parent and *STAG2* KI cell lines including the *IREB2* gene product IRP2 after infection with *IREB2* and *STAG1* sgRNAs. **(E)** Effect of sgIREB2 KO on the growth RPE1 WT and *STAG2* KO cells as determined by nuclei counting. **(F)** Effect of ferric ammonium citrate on *STAG2/IREB2* genetic interaction. Cells were infected and normalized as in (C) before 50 μM ferric ammonium citrate was added ~24 h after re-plating. **(G)** Levels of iron regulatory proteins IRP2 and ACO1 in H4 parent and *STAG2* KI cells after either no treatment or treatment with 100 μM DFO for ~16 h. **(H)** (Top) Western blot of ACO1 levels across both *STAG2+* and *STAG2–* cell lines in HAP1, H4 and RPE1 backgrounds. (Bottom) Amount of ACO1 as compared with HAP1 WT in three independent lysates. **(I)** Clonogenic growth of HAP1 WT and *STAG2* KO cells after co-transfection of either vector (pcDNA3.1) or ACO1-FLAG plasmids and *IREB2* sgRNAs. ns, not significant, *$P < 0.05$, **$P < 0.005$, ***$P < 0.0005$ in a Welch's two-tailed *t* test. ns, not significant, +$P < 0.005$, ++$P < 0.0005$ in a one-way ANOVA plus TUKEY.

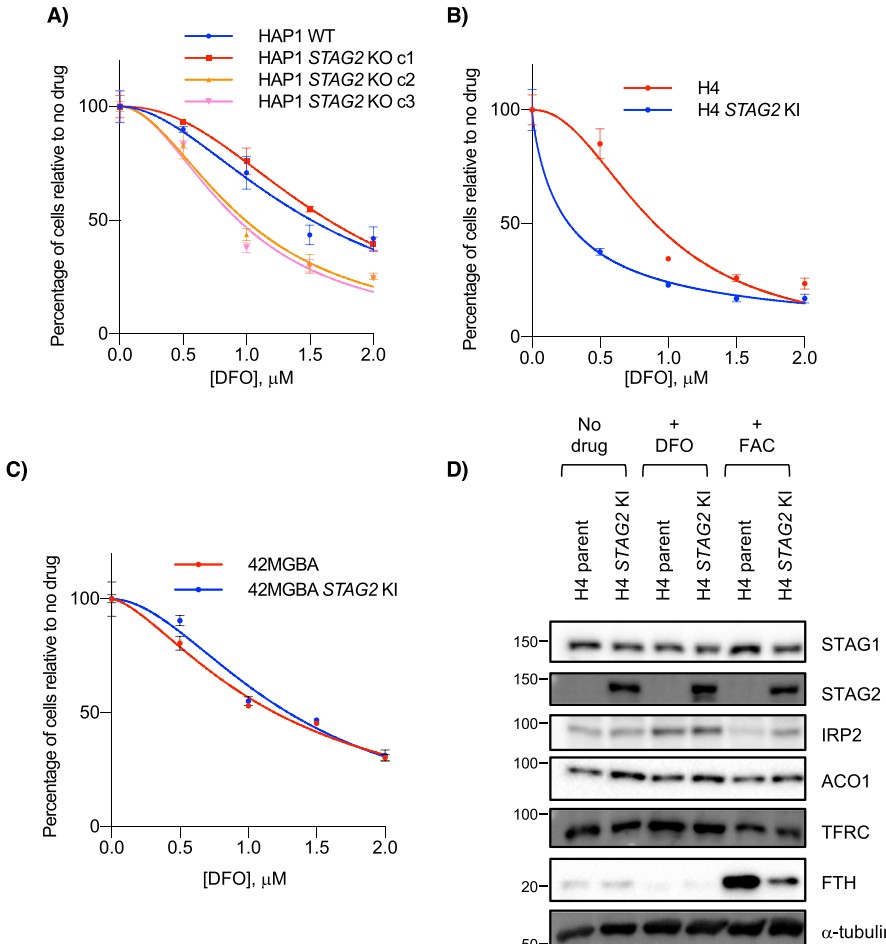

**Figure 4. *STAG2* status and iron response.**
**(A, B, C)** Cell growth of (A) HAP1, (B) H4 and (C) 42 MGBA *STAG2*+/− cells in the presence of the iron chelator DFO. **(D)** Levels of iron-related proteins including IRP targets TFRC and FTH in H4 parent and *STAG2* KI cells after no treatment or treatment with 100 µM DFO or 300 µM ferric ammonium citrate.

context-dependent interaction between *STAG1/2* and the cohesin loaders in HAP1 cells.

Further analysis of the interaction showed no obvious cell cycle changes in either WT or *STAG2* KO cells after the *MAU2* gene was deleted (Fig S5E and F). Chromatin fractionation also showed that all three core ring components (SMC1A, SMC3, and RAD21) had no detectable differences in bulk chromatin binding after loader depletion (Fig 5C). Although a previous study using *NIPBL*+/− MEF also did not observe any decrease of core cohesin components on chromatin after fractionation (Remeseiro et al, 2013), that same study and others have seen a decrease in core components in both heterozygous MEFs and HAP1 cohesin loader knockouts using ChIP and, in the case of HAP1 cells, immunofluorescence (Remeseiro et al, 2013; Haarhuis et al, 2017; Newkirk et al, 2017). Therefore, we speculate that chromatin fractionation may not be sensitive enough or our experiment may not have been long enough (only 7 d post-transfection of sgRNAs) to observe cohesin changes at the global chromatin level. Interestingly, knockout of *MAU2* in HAP1 WT cells resulted in an increase in STAG1 in lysates compared with control, a similar effect to that observed at the RNA level (Rollins et al, 2004; Kawauchi et al, 2009; Liu et al, 2009). Surprisingly, chromatin fractionations showed this increased STAG1 was mainly in the soluble fraction rather than on chromatin and seemed unlikely to be physically associated with the chromatin-bound core cohesin subunits. This up-regulation of STAG1 after *MAU2* deletion was also seen in HAP1 *STAG2* KO cells (Fig 5C).

Finding a context where cohesin loaders were non-essential was unexpected and we therefore wondered whether there were any other contexts besides HAP1 *STAG2* KO cells where cohesin loaders were not essential. Gene dependency scores in the DepMap screening dataset showed that depletion of the cohesin loader genes had a similar range of dependency scores to many of the other accessory factors of cohesin, especially when compared with the three essential core ring components (Fig S5G). Consistent with the loaders working as a heterodimer, scores of the two loader genes correlated well across all cell lines, with a very small fraction of cell lines showing positive gene dependency scores after depletion of either loader (Fig S5H). This suggests that there are other cell line contexts, albeit a very small number of them, in which loss of a cohesin loader gene showed little or no fitness defect.

Correlations in cell line essentiality profiles can suggest functional relationships between genes (Pan et al, 2018; Kim et al, 2019), so we expanded our analysis of the DepMap dataset and found three other genes, *PAXIP1* (protein name PTIP), its binding partner *PAGR1* (also known as PA1) and *PRR12*, whose gene dependency score profiles correlated well with the two loader genes and *STAG1/2*

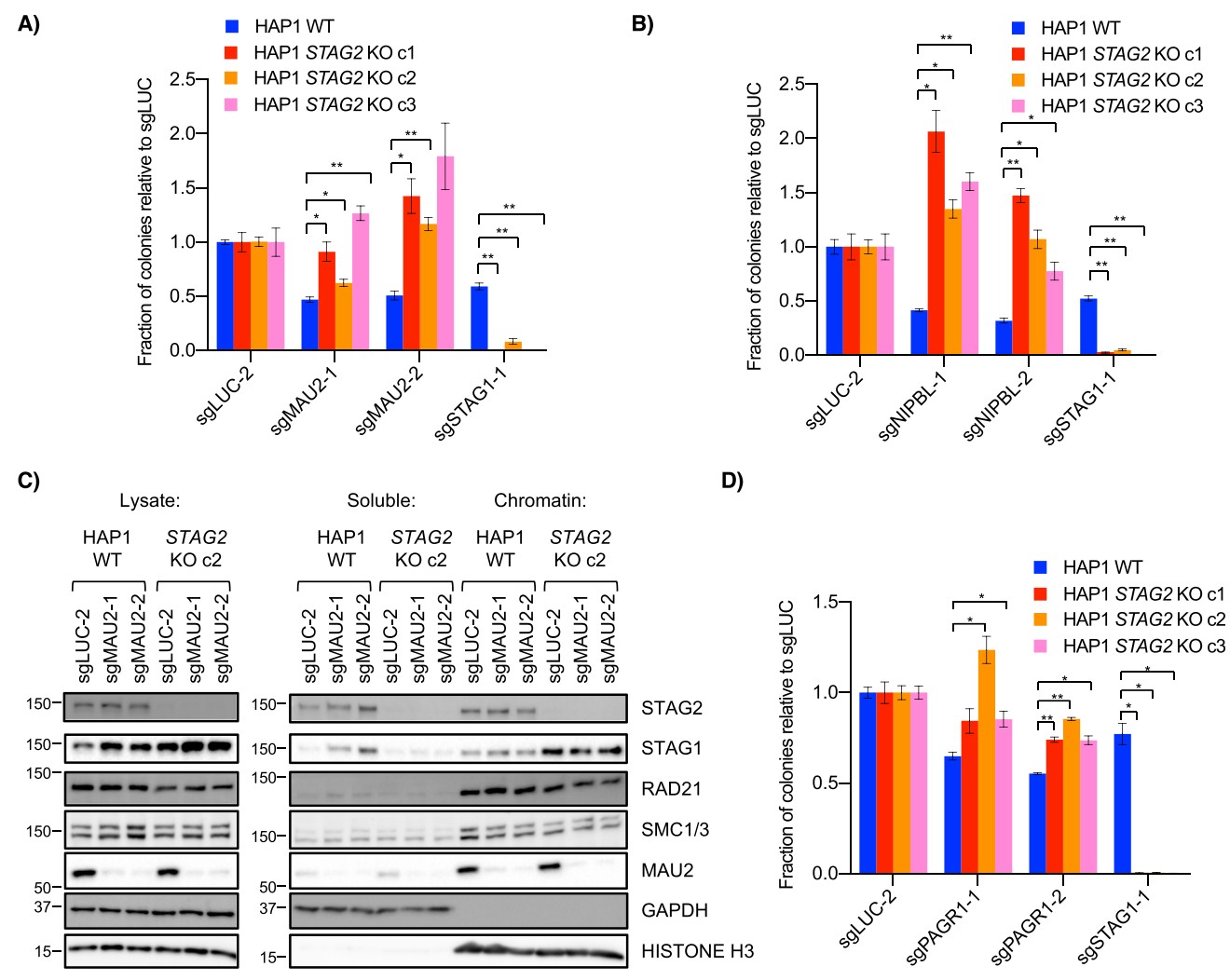

**Figure 5. Knockout of cohesin loaders in HAP1 *STAG2* KO cells.**
**(A, B, D)** Clonogenic growth of HAP1 wild-type and *STAG2* KO cells after transfection of sgRNAs for *MAU2* (A), *NIPBL* (B) or *PAGR1* (D). **(C)** Levels of cohesin proteins in lysates and soluble and chromatin fractions of HAP1 WT and *STAG2* KO cells after transfection with control and *MAU2* sgRNAs. *P < 0.05, **P < 0.005.

(Fig S5I and J). Both *PAXIP1* and *PAGR1* were candidate positive interactions in the HAP1 primary screen (qGIs of 0.882 and 0.847, respectively) and we validated *STAG2*/*PAGR1* as a positive interaction using clonogenic assays (Fig 5D).

Taken together, we found the DepMap CRISPR dataset supports the possibility of a small subset of genetic contexts, like HAP1 *STAG2* KO, where there is less dependence on cohesin loaders for cell fitness.

## Discussion

Genetic interaction profiles obtained by genome-scale CRISPR knockout screens can have functional and therapeutic implications. In recent years, however, more studies have recognized that many genetic interactions are influenced by genetic and environmental factors including cell line background, epigenetic profiles and various endogenous and exogenous stresses (Ryan

et al, 2018; Shen & Ideker, 2018). In this study, we used CRIPSR-Cas9 whole-genome screening to investigate potential genetic interactions of *STAG2*, a member of the cohesin complex that has a high rate of loss in certain types of cancer. We performed isogenic screens in three different cell backgrounds and found several points of comparison with the DepMap screening initiative, but little overlap of genetic interactions among all three backgrounds. Only one strong and conserved negative genetic interaction was found between *STAG2* and its paralog *STAG1*, a finding consistent with a recent isogenic *STAG2* screen in KBM7 cells (van der Lelij et al, 2020).

In a combinatorial CRISPR study investigating pairwise genetic interactions of 73 genes in three cell lines, (Shen et al, 2017) found that ~10% of the observed negative genetic interactions were conserved in more than one cell line, but none overlapped in all three cell lines (Shen et al, 2017). In contrast, only ~2% of *STAG2*-dependent negative genetic interactions overlapped in two or more cell line backgrounds in our screens, with one hit common in all

three backgrounds; that is, *STAG1* (Figs 1F and S1F). One reason for this lower overlap could be technical. Although a combinatorial CRISPR screen introduces both sgRNAs into a population of cells at the same time, our screens were done in lines isolated from a from a single colony where both *STAG2*-independent clonogenic effects and background-specific outgrowth adaptation could significantly decrease genetic interaction overlap.

Another reason for the lower overlap observed in our screens could be the function of STAG2 itself. The 10% negative genetic interaction overlap found by Shen et al represents an average across the 73 genes tested and it may be that certain genes simply have larger numbers of context-independent genetic interactions. For example, a recent study of PARP inhibitor-dependent gene sensitivities found that >47% reproduced in more than one cell line background; however, an independent study using an ATR inhibitor found only 9% of ATR inhibitor gene sensitivities reproduced in a second cell line (Zimmermann et al, 2018; Wang et al, 2019).

The characteristic(s) of a gene or genetic interaction that determine context-dependency are still largely unknown, but in the case of STAG2-cohesin, we offer three possible reasons for lower genetic interaction overlap between multiple contexts: (1) Although cohesin is multifunctional, not all cellular functions require the same amount of the complex. Studies in yeast and loader-deficient mammalian cells have suggested a "dosage-sensitive functional hierarchy" for cohesin where functions such as gene regulation are most sensitive to changes in cohesin levels, whereas essential mitotic functions such as sister chromatid cohesion require only a very small amount of the complex (Heidinger-Pauli et al, 2010; Newkirk et al, 2017). (2) The *STAG2* paralog, *STAG1*, can likely buffer most essential cohesin functions after *STAG2* loss. Although there is evidence for *STAG2*-specific functions in DNA repair and telomere recombination (Kong et al, 2014; Daniloski & Smith, 2017), we did not observe consistent enrichment of genes in these pathways in our primary screens under the conditions tested. (3) Other unbuffered *STAG2*-only functions appear cell type specific. *STAG2* is known to regulate cell type-specific transcription and contribute to HSPC "stemness" (Mullenders et al, 2015; Galeev et al, 2016; Kojic et al, 2018), suggesting that there may be a higher number of overlapping genetic interactions among cell backgrounds with shared differentiation programs and/or transcriptional profiles.

Both *STAG1*/*STAG2* and *IREB2*/*ACO1* are examples of paralogous synthetic lethal pairs (Smith et al, 2006; Benedetti et al, 2017; van der Lelij et al, 2017, 2020; Liu et al, 2018; Viny et al, 2019). Although we confirmed *STAG1* as a context-independent paralogous synthetic lethal interaction of *STAG2*, we also uncovered buffering between the IRP paralogs as a key factor underlying the mechanism of the *STAG2*/*IREB2* genetic interaction. Although the *STAG2*/*IREB2* genetic interaction was initially chosen because of the relationship between iron dysregulation and cancer, we found no evidence that iron played a role in the *STAG2*-negative genetic interaction with *IREB2* (Fig 3F). Instead, we linked proliferation defects in multiple isogenic contexts to lower levels of ACO1 in *STAG2* KO cells. Although more complex than the *STAG1*/*STAG2* paralogous synthetic lethal interaction, this mechanism also helps explain the context-dependency

of *STAG2*/*IREB2* as buffering of *IREB2* loss is only needed at low expression levels of ACO1 and ACO1 levels are only partly controlled by STAG2.

Although paralogous synthetic lethal pairs like *STAG1*/*STAG2* and *IREB2*/*ACO1* are interesting biologically, how the large therapeutic window of these paralogous synthetic lethal interactions can be translated into potential cancer therapeutics is an open question. Although paralogs can be easily distinguished genetically with RNAi or CRISPR, they are more difficult to distinguish pharmacologically. STAG1 and STAG2, for example, are more than 85% similar at the protein level which will make the development of paralog-specific inhibitors, especially for non-enzymatic proteins, very difficult.

The original goal of our *STAG2* screen was to look for genetic interactions common to multiple backgrounds; however, we also observed and confirmed a highly unexpected positive interaction in the HAP1 background between *STAG2* and both subunits of the cohesin loader (Figs 5 and S5). Our analysis suggested there may be some functional relationship between the cohesin loaders, STAG2 and the multifunctional PTIP/PA1 complex (Fig S5J). Exactly which function of PTIP/PA1 may be important with respect to STAG2 and cohesin function is at this point unknown. PTIP has been shown to function in both histone methylation and stalled replication fork protection through its interaction with the MLL3/4 complex and to promote DNA damage response with PA1 in a RNF8-dependent manner (Patel et al, 2007; Cho et al, 2007; Wu et al, 2009; Chaudhuri et al, 2016); however, these factors do not show an obvious correlation with the cohesin loader or *STAG2* in the DepMap data (Fig S5I). The PTIP/PA1 complex also plays a role in class switch recombination where it promotes MLL3/4-independent H3K4 methylation as well as long-range chromatin interactions (Schwab et al, 2011; Starnes et al, 2016). Furthermore, both PTIP/PA1 and cohesin loading localize mainly to open chromatin and transcriptionally active promoters, which also supports a functional association between these complexes (Lechner et al, 2000; Liu et al, 2009; Newkirk et al, 2017). Future studies that explore this relationship between cohesin loading and the PTIP/PA1 complex could provide insights into not just cohesin dynamics and function but also the developmental disease Cornelia de Lange Syndrome which is often characterized by heterozygous mutations in *NIPBL* (Sarogni et al, 2020).

Based on our screen results and those of others, it seems unlikely that a "loss-of-function" therapeutic target beyond STAG1 will be discovered that will work in all diverse cancer types with *STAG2* loss. In the future, it may be of greater benefit to determine *STAG2* dependencies in groups of cells lines that closely share a lineage, transcriptional program or a cohesin-related vulnerability. Alternatively, better overlap among *STAG2*-dependent genetic interactions might be achieved by screening for "gain-of-function" mutations in synthetic lethal partner genes or by using agents such as PARP inhibitors that have shown greater context-dependency across multiple *STAG2*-deficient and *STAG2*-mutated cell lines (McLellan et al, 2012; Yang et al, 2013; Bailey et al, 2014; Iorio et al, 2016; Mondal et al, 2019). PARP inhibitors induce genomic stress by creating DNA lesions and inhibit DNA repair, a known cohesin function.

# Materials and Methods

### Plasmids, cell lines, and antibodies

For generation of knockout lines, sgRNAs (Table S6) were cloned into pSpCas9-T2A-puro (#62988; Addgene). The puromycin resistance gene caused variable puromycin resistance in RPE1-hTERT cells, so T2A-puro was swapped for T2A-Blast using EcoRI. For all subsequent sgRNA KO experiments, sgRNAs were cloned into LentiCRISPR v2 (#52961; Addgene). An EF1α-FLAG-dCAS9-KRAB-T2A-Blast plasmid was used to express dCAS9-KRAB for all CRISPRi experiments. This construct was made by amplifying EF1α from pCRISPRia (#84832; Addgene), T2A-Blast from dCas9-VP64 Blast (#61425; Addgene) and KRAB from Lenti-dCas9-KRAB-Blast (#89567; Addgene) before sequentially cloning each product into pLV hUbC-dCas9-VP64-T2A-GFP (#53192; Addgene) using PacI/XbaI, NheI/AgeI, and NheI, respectively. CRISPRi sgRNAs were cloned into pCRISPRia. sgRNAs were verified correct by sequencing before being used. ACO1-FLAG was from GenScript.

All cell lines were grown in 10% FBS (Invitrogen) and incubated at 37°C and 5% $CO_2$. HAP1 cells were cultured in IMDM. H4 and 42 MGBA cells (gift from T Waldman) and RPE1-hTERT cells were grown in DMEM. HCT116 WT and STAG2 KO (gift from T Waldman) were grown in McCoy's medium.

Antibodies used for Western blot were as follows: from Abcam, SMC1 (ab9262), SMC3 (ab9263), α-tubulin (DM1A) (ab7291), FTH (ab75972), MAU2/Scc4 (ab183033), GAPDH (ab9485), and HISTONE H3 (ab1791); from GeneTex, STAG1 (GTX129912) and ACO1 (GTX128976); from Millipore, RAD21 (05-908); from Novus, CAS9 (NBP2-36440); from Santa Cruz, STAG2 (sc81852), IRP2 (sc33682), and TFRC (sc65582); from Sigma-Aldrich, FLAG M2 (F1804); rat-anti-tubulin was clone YOL1/34. Secondary antibodies were either goat–anti-rabbit, goat–anti-mouse, or goat–anti-rat conjugated to HRP, Alexa Fluor 488 or Cy3 (Jackson Laboratories).

### Generation of clonal knockout lines

For HAP1 and RPE1 STAG2 KO clones, parent cells were transfected with pSpCas9-T2A-Blast plasmid using XtremeGene 9 (Roche) according to manufacturer's instructions. The following day, transfected cells were selected for ~3 d using Blasticidin (Sigma-Aldrich). For HAP1 KO lines, cells were then replated at single cell density in 10 cm plates. 10–14 d after plating, colonies were picked using cloning cylinders and transferred to a 96-well dish. For RPE1 cells, cells were plated at limiting dilution in 96-well plates. All clones were passaged until they reached 10 cm density. Clones were then sequenced for a mutation near the expected target site and any expected protein knockouts were confirmed by Western blot. All parent lines and STAG2 KO clones were checked for mycoplasma before being used. HAP1 clones were also stained with propidium iodide and compared with parent cells to determine ploidy.

### CRISPR-Cas9 knockout screens

All CRISPR-Cas9 screens were performed as previously described (Aregger et al, 2019). Briefly, cells were infected with lentiviral TKOv3

library at an MOI of ~0.3 then selected the following day with puromycin (1 µg/ml for H4 cells, 2 µg/ml for HAP1 cells and 8 µg/ml for RPE1 cells) for 48 h for HAP1 and H4 cells. For RPE1 lines, cells were selected for 72 h with replacement of selection media after 24 h. After selection (i.e., T0), the cells were replated in three replicates at ~200-fold coverage of the library. Replicates were passaged every 3–4 d maintaining coverage of the sgRNA library and with three samples collected at T0 and all subsequent passages, until the infected population reached 20 doublings. Genomic DNA was purified from T0 and end point samples using Promega Wizard Genomic DNA Purification kit according to manufacturer's instructions. For each sample, sgRNA inserts were amplified from 52.5 µg of genomic DNA by a two-step PCR reaction using primers harboring Illumina TruSeq adaptors with i5 and i7 barcodes. The sequencing libraries were gel purified and sequenced on a Illumina HiSeq 2500. For the HAP1 screen, each STAG2 KO clone was run as one replicate and the three replicates were combined for dropout scoring. For RPE1 (WT and STAG2 KO c1) and H4 glioblastoma (parent and STAG2 KI) screens, a midpoint (T18 for RPE1 and T14 for H4) was also lysed and run. For the HAP1 STAG2 KO screen, $\log_2$-fold change and genetic interaction (qGI) scores were processed and calculated as in Aregger et al (2020). For RPE1 T18 and H4 T24 screens, $\log_2$-fold change and genetic interaction residuals were similarly calculated but without correction for multiple wild-type screens.

### STAG1 knockdown experiments

Cell lines were infected with the dCas9-KRAB construct. HAP1 and H4 cells were infected as in primary screens, but because of a significant growth lag, all RPE1 infections after the primary screen were performed as follows: infection with virus in 6% FBS for 5–6 h followed by recovery in virus-free medium with 10% FBS for 18–19 h. After 24 h, all infected cells were selected with Blasticidin. 7–10 d after KRAB infection, cells were collected and infected with pCRISPRia constructs. After 48 h (H4) or 72 h (RPE1) of puromycin selection, cells plated in triplicate in 96-well plates were grown in drug-free medium for 3–5 d, before being fixed with paraformaldehyde, stained with Hoechst 33342 and nuclei counted using a Cellomics LTV machine. HAP1 cells were selected for 48 h and plated in triplicate at single cell density for clonogenic assay. Plates were incubated over 7–9 d, with media changed halfway through before colonies were stained with 0.1% Crystal Violet in 95% ethanol. Colonies were then manually counted. Cells that were infected for flow cytometry were selected and grown in drug-free medium for 24–48 h before being fixed in 70% ethanol at −20°C for at least 24 h. Cells were stained with phospho-Ser10 Histone H3 (ab5176) and Goat-anti-rabbit AlexaFluor 488 (from Jackson Laboratories) for mitotic cells as well as propidium iodide (PI) for DNA content and run on a FACSCaliber flow cytometry machine. At least 10,000 cells were counted per experiment and three independent experiments were analyzed using FlowJo software.

### Cell growth experiments

For HAP1 clonogenic assays, cells were seeded in 24-well dishes and transfected with LentiCRISPR v2 plasmids using XtremeGene 9 according to manufacturer's instructions. After 48 h of puromycin selection, cells were collected and the control counted so cells could

be plated at single cell density in triplicate and assessed as described above. All other cell lines were infected with LentiCRISPRv2 containing guides targeting specific genes. After selection, cells were then grown in drug-free medium for another 24 h before cells were collected, counted, and re-plated at equal density across each cell line in triplicate in either 96-well plates for nuclei counting or six-well dishes for clonogenic assay. For drug experiments, talozoparib, olaparib and bortezomib were from Selleck, deferoxamine (DFO), and thapsigargin were from Cedarlane Labs, and camptothecin (CPT), etoposide, hydroxyurea (HU), 5-fluorouracil (5-FU), SAHA, paclitaxel and anisomycin were from Sigma-Aldrich. To generate inhibition curves, cells were plated in 96-well dishes and drug was added the following day. Cells were incubated for a further 3–4 d before nuclei were stained and counted. $IC_{50}$s were calculated using GraphPad Prism v8. For serial dilution experiments, HAP1 cells were transfected with lentiCRISPR v2, selected with puromycin and the control counted to make a cell slurry where the control would reach 80–90% confluency in 7–9 d. Cells were serially diluted 1:2 and plated in 96-well dishes and allowed to grow for 7–9 d before being stained with 0.1% crystal violet. H4 cells were infected with lentiCRISPR v2 constructs targeting specific genes, selected, counted and normalized before being serially diluted similar to HAP1 cells.

### Chromatin fractionation

HAP1 WT and KO cells were transfected and selected as above before being moved to 25-cm$^2$ flasks with multiple wells of the same transfection pooled together. Transfections grew for another 4 d before cells were collected and fractionated with a protocol similar to Méndez and Stillman (2000). Briefly, ~3–6 × 10$^6$ cells were resuspended in Buffer A1 (10 mM Hepes, pH 7.9, 10 mM KCl, 1.5 mM MgCl$_2$, 0.34 M sucrose, 10% glycerol, 1 mM DTT, and cOmplete protease inhibitors; Roche). An equal volume of Buffer A2 (Buffer A1 + 0.2% Triton X-100) was added, and cells were incubated on ice for ~7 min. Cells were spun down at 1,300$g$ for 4 min at 4°C and the supernatant (S1) removed. The pellet (P1) was washed once with Buffer A1 before being resuspended in Buffer B (3 mM EDTA, 0.2 mM EGTA, and 1 mM DTT + protease inhibitors). Cells were spun again at 1,700$g$ for 4 min at 4°C and the supernatant (S2) removed before the pellet (P2) was washed once with Buffer B and resuspended in 1 × Laemmli (2% SDS, 10% glycerol, 60 mM Tris, pH 6.8, 0.01 mg bromophenol blue) and sonicated. For lysates, ~4 × 10$^6$ cells from the same original cell slurry were lysed in 50 mM Tris–HCl, pH7.5, 150 mM NaCl, 3 mM EDTA, 10% glycerol, 1% Triton X-100, and protease inhibitors and sonicated. Both lysates and the S1 fraction were spun at ~18,000$g$ at 4°C for 15 min. The protein concentration of the lysate was determined using BCA (Thermo Fisher Scientific) and 20 $\mu$g of each lysate, or the equivalent volume of S1 (soluble) and P2 (chromatin) fractions was used for Western blot.

### Western blots

Except for those cells undergoing chromatin fractionation, all other samples for Western blot were lysed in Lysis buffer (50 mM Tris–HCl, pH 7.5, 150 mM NaCl, 10% glycerol, 1% Triton X-100, and protease inhibitors), sonicated, and debris spun down as above. Samples were normalized by protein concentration using BCA, run on SDS–PAGE gels of appropriate acrylamide concentration and transferred to

PVDF membrane (Immobilon-FL; Millipore). After probing with primary and secondary antibodies, blots were then subjected to ECL (Clarity or Clarity Max Western ECL substrate; Bio-Rad) and visualized using a Bio-Rad ChemiDoc MP Imager in the appropriate channel. For Western blot quantification, bands were quantified using the Bio-Rad ImageLab software.

### Statistics

With the exception of the bubble graphs in Fig 1, graphs and inhibition curves were generated using GraphPad Prism v8. Bubble graphs were generated with Microsoft Excel. Data points represent the mean and SEM unless otherwise indicated. $P$-values were calculated using a two-tailed, unpaired, Welch's $t$ test unless otherwise indicated.

# Supplementary Information

# Acknowledgements

We would like to thank Nigel O'Neil and Akil Hazma for discussion and review of the manuscript, Catherine Ross and Kevin Brown for help with data analysis, and Dr Todd Waldman for providing STAG2-modified cell lines. Funding was provided by a Canadian Institutes for Health Research grant to P Hieter (FDN 148449) and Canadian Institute for Advanced Research Genetic Networks Catalyst Funding to P Hieter and J Moffat.

## Author Contributions

ML Bailey: conceptualization, data curation, formal analysis, investigation, methodology, and writing—original draft, review, and editing.
D Tieu: investigation and writing—review and editing.
A Habsid: investigation and writing—review and editing.
AHY Tong: investigation and writing—review and editing.
K Chan: investigation, methodology, and writing—review and editing.
J Moffat: conceptualization, funding acquisition, investigation, and writing—review and editing.
P Hieter: conceptualization, funding acquisition, and writing—review and editing.

## Conflict of Interest Statement

The authors declare that they have no conflict of interest.

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
