## [Reviewer comments · Life Science Alliance]

Life Science Alliance

Paralogous synthetic lethality underlies genetic dependencies of the cancer-mutated gene **STAG2**

Melanie Bailey, David Tieu, Andrea Habsid, Amy Tong, Katherine Chan, Jason Moffat, and Philip Hieter

DOI: <https://doi.org/10.26508/lsa.202101083>

Corresponding author(s): Philip Hieter, University of British Columbia

Review Timeline:

Submission Date:	2021-04-01
Editorial Decision:	2021-06-21
Revision Received:	2021-08-10
Editorial Decision:	2021-08-13
Revision Received:	2021-08-14
Accepted:	2021-08-16

Transaction Report:

June 21, 2021

Re: Life Science Alliance manuscript #LSA-2021-01083

Dr. Philip Hieter
University of British Columbia
Michael Smith Laboratories
2185 East Mall
Vancouver, BC V6T 1Z4
Canada

Dear Dr. Hieter,

Thank you for submitting your manuscript entitled "Paralogous synthetic lethality underlies genetic dependencies of the cancer-mutated gene STAG2" to Life Science Alliance. The manuscript was assessed by expert reviewers, whose comments are appended to this letter. We invite you to submit a revised manuscript addressing the Reviewer comments, highlighting that additional mechanistic insights into IREB2 in this context would be important.

Thank you for this interesting contribution to Life Science Alliance. We are looking forward to receiving your revised manuscript.

Sincerely,

Eric Sawey, PhD
Executive Editor

- A letter addressing the reviewers' comments point by point.
- An editable version of the final text (.DOC or .DOCX) is needed for copyediting (no PDFs).
- High-resolution figure, supplementary figure and video files uploaded as individual files: See our detailed guidelines for preparing your production-ready images, <https://www.life-science-alliance.org/authors>
- Summary blurb (enter in submission system): A short text summarizing in a single sentence the study (max. 200 characters including spaces). This text is used in conjunction with the titles of papers, hence should be informative and complementary to the title and running title. It should describe the context and significance of the findings for a general readership; it should be written in the present tense and refer to the work in the third person. Author names should not be mentioned.

B. MANUSCRIPT ORGANIZATION AND FORMATTING:

Reviewer #1 (Comments to the Authors (Required)):

In the manuscript of Bailey et al on 'Paralogous synthetic lethality underlies genetic dependencies of the cancer-mutated gene STAG2', genome wide CRISPR screens are performed to find genetic dependencies of STAG2 KO cells. Their three most important findings are:

1. STAG1 is the sole and strongest context-independent negative genetic interaction for STAG2
2. In two out of three cellular backgrounds they find a sensitivity towards the inactivation of the iron regulatory gene IREB2
3. In HAP1 cells, they find a positive genetic interaction of STAG2 with NIPBL and MAU2, i.e. if STAG2 is lost, the cohesin loading complex becomes less essential

1. STAG1 is the sole and strongest context-independent negative genetic interaction for STAG2. The previous finding of STAG1/2 as the strongest synthetic lethal interaction in KBM7 cells genome wide (van der Lelij et al LSA 2020) is now confirmed in three additional cellular backgrounds. The significance is that it is the sole genetic interaction shared among the three cellular backgrounds, and that it is context independent. When discussing the latter, similar conclusions were made previously in bladder cancer/Ewing cells (van der Lelij et al eLife 2017), therefore should be referred to.

Besides the CRISPR screens the authors performed additional experiments: they studied the effect on viability of various drugs in their isogenic cell pairs. They find a consistent increased sensitivity in STAG2 KO only after PARPi, when compared to wt. These are interesting findings when considering STAG2 mutations as a therapeutic biomarker, which have been reported previously. As the experiments are not integrated in the story and have no further implications for/ are not discussed in the rest of the manuscript, incorporating them in the story line would be essential or it should be considered to leave them out.

2. In two out of three cellular backgrounds they find a sensitivity towards the inactivation of the iron regulatory gene IREB2.

They confirm the results of the primary screen by using independent sgRNAs and find increased sensitivity of STAG2 KO cells to IREB2 inactivation in HAP1 and H4 backgrounds, and in two additional isogenic sets of cell lines (42 MGBA and HCT116).

They perform follow up experiments to further study this interaction, by adding ferric ammonium citrate (FAC) or deferoxamine (DFO) and examine the expression of ACO1. Interestingly, the ectopic expression of ACO1 alleviates the dependence of STAG2 KO cells on IREB2. DFO treatment also increases ACO1 levels. Although they examine if DFO alone has an effect on viability in wt or STAG2 KO cells (Fig 4), it would be more interesting to see if DFO treatment can increase viability in STAG2 KO cells that are depleted for IREB2, thereby phenocopying the ACO1-FLAG experiment.

In line with this, it is interesting to see that RPE1 cells- which did not show specific sensitivity to IREB2 inactivation- show high ACO1 levels. The authors discuss that variations in ACO1 expression levels could explain the difference in the presence of STAG2/IREB2 synthetic lethality among cell lines. As this is an important point which should be elaborated, a crucial experiment would be to confirm that, as seen for a small subset in Fig 3F, ACO1 levels correlate to the sensitivity to IREB2 inactivation in a broader set of STAG2 KO cell lines. If the authors can show a mechanism that explains the variation in STAG2/IREB2 synthetic lethality among cellular backgrounds, it would greatly strengthen their findings. It would nominate ACO1 as a biomarker that can predict the effectiveness of IREB2 inactivation in STAG2 KO cells, which is valuable when considering IREB2 as a potential therapeutic target for treating STAG2 mutated cancers.

3. In HAP1 cells, the authors find a positive genetic interaction of STAG2 with NIPBL and MAU2, i.e. if STAG2 is lost, the cohesin loading complex becomes less essential.

The same holds true for depleting NIPBL or MAU2 in a STAG1 KO background. In wt HAP1 cells, depleting the cohesin loaders MAU2 or NIPBL reduces the amount of colonies grown when compared to LUC control. STAG2 KO cells however show a comparable or even increased survival after NIPBL or MAU2 depletion. In theory, this could be an interesting finding, as it implicates that the loading complex is not essential in STAG2 KO cells anymore. However, the only readout used to come to this conclusion is viability by colony survival assays, and the margins are relatively small (with high variation among clones). Suggestions for experiments to support and expand this finding are:

- To avoid any potential influence of the proliferation speed of the cells on the outcome of the experiment, it is essential to measure viability of the different genotypes after the same amount of

doublings in the control condition, instead of after the same amount of time. Alternatively, survival can be tested in a competition assay.

- In the corresponding western, no decrease in SMC1 or SMC3 on chromatin after NIPBL or MAU2 depletion is seen, which is not in line to what has been reported before (e.g. Haarhuis et al, Cell 2017). This should be interrogated/discussed.
- The cohesin and its loading should be analyzed in further detail, by e.g. immunofluorescence with pre-extraction, sister chromatid cohesion by chromosome spreads, ChIP-seq...

In addition, the authors discuss a correlation they found between PAXIP1, PAGR1 and PRR12 and NIPBL/MAU2 in their DepMap analysis. Knocking down PAGR1 also increases clonogenic survival in STAG2 KO when compared to wt, although to a lesser extent than NIPBL. An explanation or implication for this finding is missing, therefore the data could be left out, or should be included and integrated to the story.

Overall, implications for the results on IREB2, the loading complex and PAGR1, are not being discussed, likewise an outlook is missing, therefore the potential impact of the findings is not clear. Additional investigations and revisions in writing are needed to strengthen the potentially interesting results of this manuscript and increase its impact.

Additional comments:

- Figure 1B: I understand the choice for colors, as they reflect the color coding of the rest of the figures. It is however very hard to interpret which line corresponds to which cell line. Increasing the size of the shapes of the data points and giving the isogenic sets the same shape (e.g. HAP1 wt vs. STAG2 KO both a square, RPE1 both triangle etc.) will increase the easiness to read the graph.
- the difference between Fig. 1F and G and EV1F is not explained and the added value of displaying all is therefore not clear.
- Fig 2: the confirmation of STAG1/2 synthetic lethality by independent sgRNAs could be a supplemental figure.
- it would be better to replace the 42 MGBA by RPE1 cells in the main Figure 3C, and add 42 MGBA to the supplement, to have the main figures consistent with the 3 sets of cell lines.
- the difference between Fig 3D and EV3H is not clear and should be explained better if the author believes both should be in.
- Fig 4 DFO by itself has no effect, better would be to repeat this experiment in an IREB2 depleted background.
- Fig 5C: aligning the text of lysate/soluble/chromatin in the center above wt and STAG2 KO would make it easier to interpret.
- Fig EV1D: adding 'non-essential' and 'essential' to the top and lower panel resp. would make it easier to interpret.
- Fig EV3A: The schematic is very informative to understand the experiments that are done, therefore it would be good to be included in a main figure. Adding 'IRES' besides the mRNAs and 'FTH' besides the pink circle would improve the cartoon.
- Fig EV3B: the crystal violet stainings are not very clear to see, increasing contrast might help. Adding a triangle above the wells to depict the dilution series would be informative.
- Fig EV3I: this panel could go to the main figure, as it nicely confirms the knockdowns.
- Fig EV5D: it is not clear to which experiment this western corresponds, so can be left out if not being discussed.

Reviewer #2 (Comments to the Authors (Required)):

In this comprehensive and well-performed study, Bailey et al conduct a CRISPR drop-out screen in several STAG2 deficient cell lines to uncover second-site mutations that compromise cell growth/viability only upon loss of STAG2. They find that there is only one gene whose loss cannot be tolerated in a STAG2 deficient background - STAG1. Aside from STAG1, of the few SL genes that were conserved between more than one cell line, IREB2 was selected for follow-up. IREB2 encodes a protein involved in responding to and modulating available levels of iron. Despite extensive investigation that included analysis of a functional paralogue ACO1, there was no apparent mechanistic linkage between STAG2 mutation and iron metabolism, so why IREB2 is a conserved SL gene with STAG2 remains mysterious. Finally, the authors found that loss of cohesin loaders NIPBL/MAU2 are much better tolerated in a STAG2 mutant background in HAP1 cells.

The strengths of the paper is that the experiments are robust, the data are convincing, and highlight the context-dependency of genetic vulnerabilities in STAG2 mutant cells. I liked the inclusion of gene dependency scores from the DepMap screening dataset. The main weakness of the paper is one of novelty, since the main finding of STAG1 dependency has been previously reported multiple times. There are also remaining questions around the mechanisms behind IREB2 KD vulnerability and MAU2/NIPBL KD amelioration. Nevertheless, the present study is a valuable contribution to the field.

Major comments

1. "In summary, while a broader link between STAG2 and iron response was not established, STAG2-dependent and STAG2-independent effects on ACO1 expression could be responsible for the mechanism and context dependency, respectively, of the STAG2/IREB2 genetic interaction." If this statement were true then there ought to be some sort of measurable link to the iron response uncovered by experiments in STAG2 KD cells. ACO1 reduction in STAG2 KD cells is assumed to account for the SL of STAG2 with mutation of paralogue IREB2, after all. Is it possible to measure actual iron levels in the cells? Otherwise, how do the authors then account for the vulnerability? Are there co-regulated genes? Effects on cell metabolism?

2. "Both PTIP/PA1 and cohesin loading localize mainly to open chromatin and transcriptionally active promoters, further supporting the possibility of a functional association between these complexes" - PTIP/PA1 are DNA damage response proteins, and their absence might help damaged cells avoid the DNA damage checkpoint. A possible alternative interpretation is that DNA damage is sensitised in STAG2 KD, and DNA damage repair is bypassed when PTIP/PA1 are KD and cells divide faster as a result. I suggest performing staining with gamma H2AX to explore this possibility, and/or perhaps test cell status with checkpoint antibodies.

3. Discussion/Introduction - it would be fairer of the authors to be more inclusive of the extensive prior literature on SL with STAG2 KD, these are rather selectively cited in this manuscript.

Previous studies directly related to STAG2 being SL with STAG1:

Van der Lelij 2020 and 2017: PMID: 32467316, PMID: 28691904

Viny 2019: PMID: 31495782

Benedetti 2017: PMID: 28430577

Liu: PMID 2018: 29649003

Minor comments

Annoying not to have page or line numbers.

Fig EV2 - panel order on the page is odd, better to have letters A and B side by side and letter C below.

Fig 2A - why do the wild type cells grow better when STAG1 is knocked down?

"suggested a role for both STAG1 and STAG2 in the chromatin localization of cohesin" - should really cite the Losada 2018 paper for this statement doi: 10.1038/s41594-018-0070-4 and Pherson et al 2019: PMID: 30796039

Reviewer #3 (Comments to the Authors (Required)):

In this paper, Bailey et al use CRISPR Cas9 whole genome libraries to screen for genetic interactions with Stag2, one of the core members of cohesion complex frequently mutated in several human cancers. Authors perform this screening in three different cell lines reasoning that finding therapeutic targets common in several cell backgrounds can have the advantage of become useful not only across multiple cancer types but also "in the context of tumor heterogeneity". Authors found the Stag2 paralog Stag1 as the only context independent negative interactor. This result confirms previous data obtained by many authors in different cell types (Mondal et al., 2019; Van der Lelij et al, 2017 and 2020; Benedetti et al, Oncotarget, 2017-this important reference is missing in the manuscript-) and highlights, as discussed by the authors, that the nature and cell type-specific functions of cohesion complex hinders the identification of Stag2 negative genetic interactors (beyond Stag1) common in a broad range of cell types. In addition, they describe a negative genetic interaction between Stag2 and IREB2, an iron regulatory gene, shared by H4 and HAP1 cells, and a positive interaction with cohesion loading complex (Mau2 and Nipbl), specifically in HAP1 cells, where Mau2 or Nipbl loss confers a positive advantage for Stag2 KO cells. Overall, the manuscript is well written and deals with an important issue, which is the identification of genetic interactors of one of the genes more frequently mutated in human cancer. However, in my opinion, this work does not provide strong-enough contribution to the current knowledge in the field. Many important experiments lack essential controls and some of the statements and conclusions are not sufficiently supported.

Specific comments are the following:

Figure 1

- Rational behind the selection of these three cell lines should be clearly explained.
- Rational behind the selection of these three cell lines should be clearly explained different in all cell lines in the absence of Stag2. It is not clear to me if this starting point corresponds to a low dose of the drug or if it is the "untreated" condition.
- Statistical analysis of the genetic interactions should be more clearly explained: for example, it is confusing to me why two different FDR cutoffs are shown (<0.2 and <0.5 in EV1F and Figure 1F respectively) and how this conciliates with the cut off based on qGI. This is important considering that authors selected IREB2 negative interaction with Stag2 in HAP1 and H4 cells for the subsequent studies, when this candidate passes over the FDR<0.2 threshold.

Figure 2

- Authors show decrease in cell growth and increased mitotic index in Stag2 KO cells depleted from Stag1. However, it is intriguing to me the low level of cell death (in the absence of cohesin complex) observed in RPE and H4 cells when compared with HAP1 cells. This might be related with differences in the remaining chromatin bound cohesin in the three different cell lines, which should be shown (authors only show the remaining total protein levels of Stag1 upon infection in RPE1 cells, EV2A).

Figure 3 and 4

- WB showing IREB2 depletion levels in the different cell lines should be shown.
- The sentence "Interestingly, we did not see a negative genetic interaction between STAG2 and IREB2 in RPE1 cell lines, consistent with the primary screen data. Therefore, we conclude that STAG2/IREB2 is a negative genetic interaction in many, but not all cellular contexts" is an overstatement. Authors see this interaction in two out of three cell lines, and given this limited number, I don't think it is possible to make such general conclusions. It would be interesting to explore the genetic interactions between this IREB2 pathway and Stag2 in different cellular contexts, such as those recently available in the DepMap portal from Kim Stegmaier group (Dharia et al., Nature Genetics, 2021).

Figure 5

I find intriguing the positive genetic interaction of cohesion loading complex (Mau2 and Nipbl) members and Stag2 which seem to be exclusive in HAP1 cells. Did authors check if this interaction does not take place in any of the other cell lines in the study?

Based on correlations in cell line essentiality profiles, authors propose PTIP/PAI proteins as possible genetic interactors. Furthermore, they indicate that since both PTIP/PAI proteins and cohesin complex localize to open chromatin and active promoters they could be functionally related. It would be interesting to compare their genome-wide distribution by ChIP-seq (with already available data) to support this hypothesis.

Dear Dr. Sawey,

Please find submitted our revised manuscript entitled “Paralogous synthetic lethality underlies genetic dependencies of the cancer-mutated gene STAG2”. We appreciate the comments that the reviewers have made regarding the original submission and have addressed them in a point-by-point response below. We believe our paper meets the intended scope of the Life Science Alliance journal to publish “quality research papers on new results including negative and refuting data, as well as important confirmatory data, resources such as datasets, screens, and new methods from the full spectrum of life science and biomedical research.” In addition to our important findings, we feel the reagents and CRISPR screen datasets are of high quality and will be useful to the field. We have added data suggested by the reviewers in response to several points to strengthen the paper and we believe that it is now ready for publication.

Reviewer #1 (Comments to the Authors (Required)):

In the manuscript of Bailey et al on 'Paralogous synthetic lethality underlies genetic dependencies of the cancer-mutated gene STAG2', genome wide CRISPR screens are performed to find genetic dependencies of STAG2 KO cells. Their three most important findings are:

1. STAG1 is the sole and strongest context-independent negative genetic interaction for STAG2
2. In two out of three cellular backgrounds they find a sensitivity towards the inactivation of the iron regulatory gene IREB2
3. In HAP1 cells, they find a positive genetic interaction of STAG2 with NIPBL and MAU2, i.e. if STAG2 is lost, the cohesin loading complex becomes less essential

1. STAG1 is the sole and strongest context-independent negative genetic interaction for STAG2.

1-1) The previous finding of STAG1/2 as the strongest synthetic lethal interaction in KBM7 cells genome wide (van der Lelij et al LSA 2020) is now confirmed in three additional cellular backgrounds. The significance is that it is the sole genetic interaction shared among the three cellular backgrounds, and that it is context independent. When discussing the latter, similar conclusions were made previously in bladder cancer/Ewing cells (van der Lelij et al eLife 2017), therefore should be referred to.

We have added these findings to the manuscript on pg 7 (lines 159-160). The final concluding sentence now reads:

“This suggests that, compared with *STAG2*⁺ cells, *STAG2*⁻ cells are more dependent on *STAG1* on a large, context-independent scale, a finding consistent with previous knockdown of *STAG1* in bladder cancer and Ewing’s cell line panels (van der Lelij *et al*, 2017).”

1-2) Besides the CRISPR screens the authors performed additional experiments: they studied the effect on viability of various drugs in their isogenic cell pairs. They find a consistent increased sensitivity in STAG2 KO only after PARPi, when compared to wt. These are interesting findings when considering STAG2 mutations as a therapeutic biomarker, which have been reported previously. As the experiments are not integrated in the story and have no further implications for/ are not discussed in the rest of the manuscript, incorporating them in the story line would be essential or it should be considered to leave them out.

The intended purpose of the initial drug panel was to better character the *STAG2* KO lines at the phenotypic level (in addition to characterizing their genomic mutation and cohesin protein levels) before

the lines were used for screening. We can see that this was not clear and have re-written the paragraph on pg. 5-6 (lines 112-120) to try to better reflect our intentions.

2. In two out of three cellular backgrounds they find a sensitivity towards the inactivation of the iron regulatory gene IREB2.

1-3) They confirm the results of the primary screen by using independent sgRNAs and find increased sensitivity of STAG2 KO cells to IREB2 inactivation in HAP1 and H4 backgrounds, and in two additional isogenic sets of cell lines (42 MGBA and HCT116).

They perform follow up experiments to further study this interaction, by adding ferric ammonium citrate (FAC) or deferoxamine (DFO) and examine the expression of ACO1. Interestingly, the ectopic expression of ACO1 alleviates the dependence of STAG2 KO cells on IREB2. DFO treatment also increases ACO1 levels. Although they examine if DFO alone has an effect on viability in wt or STAG2 KO cells (Fig 4), it would be more interesting to see if DFO treatment can increase viability in STAG2 KO cells that are depleted for IREB2, thereby phenocopying the ACO1-FLAG experiment.

DFO treatment does not increase ACO1 levels, only IREB2/IRP2 levels. This is shown in Fig. 4D and EV4D and is consistent with previous literature on the post-translation role of iron in regulation of ACO1 (eg. PMID 25306858). We understand that this can be confusing and have moved Fig. S3A to Fig. 3A as suggested in point 1-17 and also mentioned this in the text of pg 8 (lines 171-173).

As DFO does not increase ACO1 levels, we would not necessarily predict it would rescue the STAG2/IREB2 genetic interaction. We used FAC in determining the role of iron in the IREB2/STAG2 genetic interaction because it is far less toxic than DFO. We have examined the interaction with a sub-lethal dose of DFO, and consistent with FAC results in Fig. 3F, found no effect on the strength of the genetic interaction. This DFO data is now Fig. S3J.

1-4) In line with this, it is interesting to see that RPE1 cells- which did not show specific sensitivity to IREB2 inactivation- show high ACO1 levels. The authors discuss that variations in ACO1 expression levels could explain the difference in the presence of STAG2/IREB2 synthetic lethality among cell lines. As this is an important point which should be elaborated, a crucial experiment would be to confirm that, as seen for a small subset in Fig 3F, ACO1 levels correlate to the sensitivity to IREB2 inactivation in a broader set of STAG2 KO cell lines. If the authors can show a mechanism that explains the variation in STAG2/IREB2 synthetic lethality among cellular backgrounds, it would greatly strengthen their findings. It would nominate ACO1 as a biomarker that can predict the effectiveness of IREB2 inactivation in STAG2 KO cells, which is valuable when considering IREB2 as a potential therapeutic target for treating STAG2 mutated cancers.

We explored lower ACO1 levels as a mechanism for the STAG2/IREB2 genetic interaction by performing a rescue experiment in HAP1 isogenic cells expressing exogenous ACO1. This showed that differences in ACO1 levels are responsible for the STAG2/IREB2 interaction (Fig. 3I). However, DepMap data from different cell lines show that lines with lower expression of ACO1 are more affected by IREB2 KO regardless of STAG2 status below a certain threshold (Fig. S3K). This inconsistency is likely explained by the fact that we used isogenic cell lines in our experiments and because differences in absolute ACO1 expression across various backgrounds likely involve factors beyond STAG2. This is consistent with the role of STAG2 and cohesin in affecting only small non-absolute changes in gene expression (eg. PMID 19468298, 28855971, 29867216). Our data suggest that the ACO1/IREB2 paralogous SL mechanism underlies the STAG2/IREB2 NGI in isogenic lines which we have tried to emphasize in the text but are uncomfortable calling ACO1 a biomarker or IREB2 a therapeutic target for general STAG2 KO at this time.

3. In HAP1 cells, the authors find a positive genetic interaction of STAG2 with NIPBL and MAU2, i.e. if

STAG2 is lost, the cohesin loading complex becomes less essential.

The same holds true for depleting NIPBL or MAU2 in a STAG1 KO background. In wt HAP1 cells, depleting the cohesin loaders MAU2 or NIPBL reduces the amount of colonies grown when compared to LUC control. STAG2 KO cells however show a comparable or even increased survival after NIPBL or MAU2 depletion. In theory, this could be an interesting finding, as it implicates that the loading complex is not essential in STAG2 KO cells anymore. However, the only readout used to come to this conclusion is viability by colony survival assays, and the margins are relatively small (with high variation among clones). Suggestions for experiments to support and expand this finding are:

1-5) - To avoid any potential influence of the proliferation speed of the cells on the outcome of the experiment, it is essential to measure viability of the different genotypes after the same amount of doublings in the control condition, instead of after the same amount of time. Alternatively, survival can be tested in a competition assay.

While we understand the concerns raised about differential growth rates among cell lines, we feel that the clonogenic assays used to validate the HAP1 genetic interaction helps mitigate the proliferation concern as in this assay, we count all colonies that are visible on the plate regardless of their size or the number of cells in them. The 7-9 day time frame of the experiment was chosen to ensure all lines formed visible colonies in the control condition while sgSTAG1 was also included in these experiments as a known negative genetic interaction.

To further address the problem of variation among clones, we have included an experiment where both STAG2 and MAU2 sgRNAs were co-transfected. This experiment was performed in the HAP1 WT background and also shows a positive interaction between STAG2 and MAU2. It is included as Fig. S5D and mentioned in the text on pg. 11 (lines 252-253).

1-6) - In the corresponding western, no decrease in SMC1 or SMC3 on chromatin after NIPBL or MAU2 depletion is seen, which is not in line to what has been reported before (e.g. Haarhuis et al, Cell 2017). This should be interrogated/discussed.

We have expanded the paragraph on the chromatin fractionation experiment to better explain our results using this bulk assay. The added portion on pg. 11-12 (lines 259-265) now reads:

“Although a previous study using NIPBL^{+/−} MEFs also did not observe any decrease of core cohesin components on chromatin after fractionation (Remeseiro *et al*, 2013), that same study and others have seen a decrease in core components in both heterozygous MEFs and HAP1 cohesin loader knockouts using ChIP and, in the case of HAP1 cells, immunofluorescence (Remeseiro *et al*, 2013; Newkirk *et al*, 2017; Haarhuis *et al*, 2017). Therefore, we speculate that chromatin fractionation may not be sensitive enough or our experiment may not have been long enough (only 7 days post-transfection of sgRNAs) to observe cohesin changes at the global chromatin level.”

1-7) - The cohesin and its loading should be analyzed in further detail, by e.g. immunofluorescence with pre-extraction, sister chromatid cohesion by chromosome spreads, ChIP-seq...

While these experiments would be interesting, we feel that they are beyond the scope of the paper.

1-8) In addition, the authors discuss a correlation they found between PAXIP1, PAGR1 and PRR12 and NIPBL/MAU2 in their DepMap analysis. Knocking down PAGR1 also increases clonogenic survival in STAG2 KO when compared to wt, although to a lesser extent than NIPBL. An explanation or implication for this finding is missing, therefore the data could be left out, or should be included and integrated to the story.

We have added insights into this interaction into the Discussion of pg. 15-16 (lines 352-369) in response to reviewer #2 (point 2-2).

Overall, implications for the results on IREB2, the loading complex and PAGR1, are not being discussed, likewise an outlook is missing, therefore the potential impact of the findings is not clear. Additional investigations and revisions in writing are needed to strengthen the potentially interesting results of this manuscript and increase its impact.

Additional comments:

1-9) - Figure 1B: I understand the choice for colors, as they reflect the color coding of the rest of the figures. It is however very hard to interpret which line corresponds to which cell line. Increasing the size of the shapes of the data points and giving the isogenic sets the same shape (e.g. HAP1 wt vs. STAG2 KO both a square, RPE1 both triangle etc.) will increase the easiness to read the graph.

The figure has been updated to reflect these changes

1-10) - the difference between Fig. 1F and G and EV1F is not explained and the added value of displaying all is therefore not clear.

Fig. 1F is the genetic interaction overlap among the lines based on a statistical cut-off while Fig. 1G is the interaction overlap based on a cut-off for genetic interaction strength. To try to make things clearer, we have eliminated the previous Fig. S1F (which was just another statistical cut-off) and moved the previous Fig. 1F to the supplemental. We have also added a sentence to the results section on pg. 6 (lines 129-130) to explain the difference between the new Fig. 1F and EV1F.

1-11) - Fig 2: the confirmation of STAG1/2 synthetic lethality by independent sgRNAs could be a supplemental figure.

We believe the alignment of the STAG1/2 synthetic lethality with the increased mitotic index in each of the cell backgrounds is informative and have left the top panels in Fig. 2.

1-12) - it would be better to replace the 42 MGBA by RPE1 cells in the main Figure 3C, and add 42 MGBA to the supplement, to have the main figures consistent with the 3 sets of cell lines.

We have switched these two panels

1-13) - the difference between Fig 3D and EV3H is not clear and should be explained better if the author believes both should be in.

We have replaced EV3H (the acute FAC panel) with the chronic DFO panel requested above (point 1-3). It is now Fig. S3J.

1-14) - Fig 4 DFO by itself has no effect, better would be to repeat this experiment in an IREB2 depleted background.

We have included an experiment with DFO as Fig. S3J and explained in reviewer's point 1-3.

1-15) - Fig 5C: aligning the text of lysate/soluble/chromatin in the center above wt and STAG2 KO would make it easier to interpret.

We have centred this text.

1-16) - Fig EV1D: adding 'non-essential' and 'essential' to the top and lower panel resp. would make it easier to interpret.

We have added this text to the figure.

1-17) - Fig EV3A: The schematic is very informative to understand the experiments that are done, therefore it would be good to be included in a main figure. Adding 'IREs' besides the mRNAs and 'FTH' besides the pink circle would improve the cartoon.

We have moved the schematic and added the text to the figure as requested

1-18) - Fig EV3B: the crystal violet stainings are not very clear to see, increasing contrast might help. Adding a triangle above the wells to depict the dilution series would be informative.

We have tried to make the pictures clearer and added the triangle.

1-19) - Fig EV3I: this panel could go to the main figure, as it nicely confirms the knockdowns.

This panel has been moved to Fig. 3D.

1-20) - Fig EV5D: it is not clear to which experiment this western corresponds, so can be left out if not

being discussed.

This panel has been removed.

Reviewer #2 (Comments to the Authors (Required)):

In this comprehensive and well-performed study, Bailey et al conduct a CRISPR drop-out screen in several STAG2 deficient cell lines to uncover second-site mutations that compromise cell growth/viability only upon loss of STAG2. They find that there is only one gene whose loss cannot be tolerated in a STAG2 deficient background - STAG1. Aside from STAG1, of the few SL genes that were conserved between more than one cell line, IREB2 was selected for follow-up. IREB2 encodes a protein involved in responding to and modulating available levels of iron. Despite extensive investigation that included analysis of a functional paralogue ACO1, there was no apparent mechanistic linkage between STAG2 mutation and iron metabolism, so why IREB2 is a conserved SL gene with STAG2 remains mysterious. Finally, the authors found that loss of cohesin loaders NIPBL/MAU2 are much better tolerated in a STAG2 mutant background in HAP1 cells.

The strengths of the paper is that the experiments are robust, the data are convincing, and highlight the context-dependency of genetic vulnerabilities in STAG2 mutant cells. I liked the inclusion of gene dependency scores from the DepMap screening dataset. The main weakness of the paper is one of novelty, since the main finding of STAG1 dependency has been previously reported multiple times. There are also remaining questions around the mechanisms behind IREB2 KD vulnerability and MAU2/NIPBL KD amelioration. Nevertheless, the present study is a valuable contribution to the field.

Major comments

2-1. "In summary, while a broader link between STAG2 and iron response was not established, STAG2-dependent and STAG2-independent effects on ACO1 expression could be responsible for the mechanism and context dependency, respectively, of the STAG2/IREB2 genetic interaction." If this statement were true then there ought to be some sort of measurable link to the iron response uncovered by experiments in STAG2 KD cells. ACO1 reduction in STAG2 KD cells is assumed to account for the SL of STAG2 with mutation of paralogue IREB2, after all. Is it possible to measure actual iron levels in the cells? Otherwise, how do the authors then account for the vulnerability? Are there co-regulated genes? Effects on cell metabolism?

We realize this statement was confusing and have expanded it to better explain our conclusions in relation to the literature. Mainly that IREB2 and ACO1 are highly compensatory in cells and that IREB2 is suggested to be the more important regulator in most tissues. In STAG2 KO cells, the presence of IREB2 heavily buffers the decrease in ACO1. Viability phenotypes only occur after IREB2 is removed in STAG2 KO cells likely because ACO1 is now the dominant paralog and cells are more susceptible to subtle changes in levels. The re-written concluding statements and references can be found on pg. 10 (lines 226-235).

2-2. "Both PTIP/PA1 and cohesin loading localize mainly to open chromatin and transcriptionally active promoters, further supporting the possibility of a functional association between these complexes" - PTIP/PA1 are DNA damage response proteins, and their absence might help damaged cells avoid the DNA damage checkpoint. A possible alternative interpretation is that DNA damage is sensitised in

STAG2 KD, and DNA damage repair is bypassed when PTIP/PA1 are KD and cells divide faster as a result. I suggest performing staining with gamma H2AX to explore this possibility, and/or perhaps test cell status with checkpoint antibodies.

We acknowledge that there are several functions where cohesin and the PTIP/PA1 complexes overlap including transcriptional activation, DNA damage and long-range chromatin interactions. Increases in DNA damage foci and issues with cell cycle checkpoints have been shown previously for several STAG2-depleted lines including H4 and RPE1 (eg. PMID 24356817, 30975996). Our STAG2 KO lines also show varying sensitivities to different DNA damaging agents (eg. Fig. S1B). As H4 and RPE1 cell lines do not show an interaction between STAG2 and the cohesin loaders (now added as Fig. S5A, B) but HAP1 cells do and all of these have some phenotypes associated with DNA damage, it seems unlikely the mechanism behind the HAP1 STAG2/MAU2 interaction is a global DNA damage mechanism. An analysis of specific DNA repair pathways could provide more insight; however, we feel this is beyond the scope of this paper.

We have addressed the potential roles of PTIP/PA1 and STAG2 and cohesin loading in a paragraph in the Discussion on pg. 15-16 (lines 352-369).

2-3. Discussion/Introduction - it would be fairer of the authors to be more inclusive of the extensive prior literature on SL with STAG2 KD, these are rather selectively cited in this manuscript.

Previous studies directly related to STAG2 being SL with STAG1:

Van der Lelij 2020 and 2017: PMID: 32467316, PMID: 28691904

Viny 2019: PMID: 31495782

Benedetti 2017: PMID: 28430577

Liu: PMID 2018: 29649003

We have added these references to the introduction on pg. 3 (lines 67-68), the results on pg. 6 (lines 138-139) and the discussion on pg. 15 (lines 335-336).

Minor comments

2-4) Annoying not to have page or line numbers.

Page and line numbers have been added.

2-5) Fig EV2 - panel order on the page is odd, better to have letters A and B side by side and letter C below.

These panels have been rearranged.

2-6) Fig 2A - why do the wild type cells grow better when STAG1 is knocked down?

Two other sgRNA's for STAG1 KD do not grow statistically better than WT (one of these is not included in Fig. 2A). As the sgRNA displayed a similar MI and cell cycle profile in HAP1 WT cells, we suspect there may be some other STAG1-independent effect of this sgRNA in HAP1 cells that we chose not to pursue at this time.

2-7) "suggested a role for both STAG1 and STAG2 in the chromatin localization of cohesin" - should really cite the Losada 2018 paper for this statement doi: 10.1038/s41594-018-0070-4 and Pherson et al 2019: PMID: 30796039

These references have been added.

Reviewer #3 (Comments to the Authors (Required)):

In this paper, Bailey et al use CRISPR Cas9 whole genome libraries to screen for genetic interactions with

Stag2, one of the core members of cohesion complex frequently mutated in several human cancers. Authors perform this screening in three different cell lines reasoning that finding therapeutic targets common in several cell backgrounds can have the advantage of become useful not only across multiple cancer types but also "in the context of tumor heterogeneity". Authors found the Stag2 paralog Stag1 as the only context independent negative interactor. This result confirms previous data obtained by many authors in different cell types (Mondal et al., 2019; Van der Lelij et al, 2017 and 2020; Benedetti et al, Oncotarget, 2017-this important reference is missing in the manuscript-) and highlights, as discussed by the authors, that the nature and cell type-specific functions of cohesion complex hinders the identification of Stag2 negative genetic interactors (beyond Stag1) common in a broad range of cell types. In addition, they describe a negative genetic interaction between Stag2 and IREB2, an iron regulatory gene, shared by H4 and HAP1 cells, and a positive interaction with cohesion loading complex (Mau2 and Nipbl), specifically in HAP1 cells, where Mau2 or Nipbl loss confers a positive advantage for Stag2 KO cells.

Overall, the manuscript is well written and deals with an important issue, which is the identification of genetic interactors of one of the genes more frequently mutated in human cancer. However, in my opinion, this work does not provide strong-enough contribution to the current knowledge in the field. Many important experiments lack essential controls and some of the statements and conclusions are not sufficiently supported.

Specific comments are the following:

Figure 1

- Rational behind the selection of these three cell lines should be clearly explained.

3-1) We have expanded the first paragraph of the results on pg. 5 (lines 101-103, 107-108) to better explain why these lines were chosen.

- Rational behind the selection of these three cell lines should be clearly explained different in all cell lines in the absence of Stag2. It is not clear to me if this starting point corresponds to a low dose of the drug or if it is the "untreated" condition.

3-2) We believe the reviewer is referring to Fig. 1B. All lines were normalized to their own no drug control and a line has been added to the figure legend to reflect this.

- Statistical analysis of the genetic interactions should be more clearly explained: for example, it is confusing to me why two different FDR cutoffs are shown (<0.2 and <0.5 in EV1F and Figure 1F respectively) and how this conciliates with the cut off based on qGI. This is important considering that authors selected IREB2 negative interaction with Stag2 in HAP1 and H4 cells for the subsequent studies, when this candidate passes over the $FDR < 0.2$ threshold.

3-3) As per Reviewer #1 (point 1-10), we have removed Fig. S1F and moved Fig. F to this position. We have also added a sentence to the results section to better explain these cut-offs.

Figure 2

- Authors show decrease in cell growth and increased mitotic index in Stag2 KO cells depleted from Stag1. However, it is intriguing to me the low level of cell death (in the absence of cohesin complex) observed in RPE and H4 cells when compared with HAP1 cells. This might be related with differences in the remaining chromatin bound cohesin in the three different cell lines, which should be shown (authors only show the remaining total protein levels of Stag1 upon infection in RPE1 cells, EV2A).

3-4) This higher levels of cell death in flow cytometry profiles is unlikely to be related directly to cohesin, but to the backgrounds of the lines themselves. HAP1 haploid cells are known to have increased mitotic defects, slippage and mitotic death (PMID 29712735) which are not known phenotypes of RPE1 or H4 cells. HAP1 cells also have impaired p53 function (PMID 24055153) while our RPE1 cells are p53 positive

(PMID 31464370). To our knowledge, the p53 function of H4 cells is not known. We have added a sentence to pg. 7 (lines 149-152) of the manuscript to better explain this difference.

Figure 3 and 4

- WB showing IREB2 depletion levels in the different cell lines should be shown.

3-5) We have provided WBs for HAP1 and RPE1 cells in Fig. S3C, H. The H4 WB panel has been moved to Fig. 3D.

• The sentence "Interestingly, we did not see a negative genetic interaction between STAG2 and IREB2 in RPE1 cell lines, consistent with the primary screen data. Therefore, we conclude that STAG2/IREB2 is a negative genetic interaction in many, but not all cellular contexts" is an overstatement. Authors see this interaction in two out of three cell lines, and given this limited number, I don't think it is possible to make such general conclusions. It would be interesting to explore the genetic interactions between this IREB2 pathway and Stag2 in different cellular contexts, such as those recently available in the DepMap portal from Kim Stegmaier group (Dharia et al., Nature Genetics, 2021).

3-6) This sentence was not meant to be a generalization but rather to refer specifically to the isogenic contexts in which experiments were performed. We have corrected the text to reflect this on pg. 8 (lines 189-190). We have also added a figure of the most recent DepMap data showing IREB2 gene effects scores sorted by STAG2 mutation in various contexts. This is now Fig. S3I and is mentioned in the results on pg. 8 (lines 186-189).

Figure 5

I find intriguing the positive genetic interaction of cohesion loading complex (Mau2 and Nipbl) members and Stag2 which seem to be exclusive in HAP1 cells. Did authors check if this interaction does not take place in any of the other cell lines in the study?

3-7) We performed these experiments in H4 and RPE1 cells which are now present in the manuscript as Fig. S5A, B.

Based on correlations in cell line essentiality profiles, authors propose PTIP/PAI proteins as possible genetic interactors. Furthermore, they indicate that since both PTIP/PAI proteins and cohesin complex localize to open chromatin and active promoters they could be functionally related. It would be interesting to compare their genome-wide distribution by ChIP-seq (with already available data) to support this hypothesis.

3-8) We feel that this analysis is beyond the scope of this paper. In this regard, in response to reviewer 2 (point 2-2), we have added a lengthier explanation of the various points of *STAG2/PAXIP1* functional overlap to the Discussion on pg. 15-16 (lines 352-369).

August 13, 2021

RE: Life Science Alliance Manuscript #LSA-2021-01083R

Dr. Philip Hieter
University of British Columbia
Michael Smith Laboratories
2185 East Mall
Vancouver, BC V6T 1Z4
Canada

Dear Dr. Hieter,

Thank you for submitting your revised manuscript entitled "Paralogous synthetic lethality underlies genetic dependencies of the cancer-mutated gene STAG2". We would be happy to publish your paper in Life Science Alliance pending final revisions necessary to meet our formatting guidelines.

- please label the abstract
- please add the Twitter handle of your host institute/organization as well as your own or/and one of the authors in our system
- please incorporate the Supplemental Materials into the main Materials & Methods section
- please upload Table S1 and S6 as individual files

FIGURE CHECK:

- please indicate molecular weights next to each protein blot

LSA now encourages authors to provide a 30-60 second video where the study is briefly explained. We will use these videos on social media to promote the published paper and the presenting author. Corresponding or first-authors are welcome to submit the video. Please submit only one video per manuscript. The video can be emailed to contact@life-science-alliance.org

A. FINAL FILES:

B. MANUSCRIPT ORGANIZATION AND FORMATTING:

Sincerely,

Eric Sawey, PhD
Executive Editor
Life Science Alliance

<http://www.lsajournal.org>

August 16, 2021

RE: Life Science Alliance Manuscript #LSA-2021-01083RR

Dr. Philip Hieter
University of British Columbia
Michael Smith Laboratories
2185 East Mall
Vancouver, BC V6T 1Z4
Canada

Dear Dr. Hieter,

Thank you for submitting your Research Article entitled "Paralogous synthetic lethality underlies genetic dependencies of the cancer-mutated gene STAG2". It is a pleasure to let you know that your manuscript is now accepted for publication in Life Science Alliance. Congratulations on this interesting work.

DISTRIBUTION OF MATERIALS:

Again, congratulations on a very nice paper. I hope you found the review process to be constructive and are pleased with how the manuscript was handled editorially. We look forward to future exciting submissions from your lab.

Sincerely,
